# Regional Beryllium-10 production rate for the mid-elevation mountainous regions in central Europe, deduced from a multi-method study of moraines and lake sediments in the Black Forest

Felix Martin Hofmann[1], Claire Rambeau[2], Lukas Gegg[1], Melanie Schulz[1], Martin Steiner[1], Alexander Fülling[1], Laëtitia Léanni[3], Frank Preusser[1], ASTER Team[3,*]

[1]Institute of Earth and Environmental Sciences, University of Freiburg, Freiburg, 79104, Germany
[2]Laboratoire Image, Ville, Environnement (LIVE UMR 7362), CNRS/Université de Strasbourg/ENGEES, France
[3]Aix-Marseille Université, CNRS, IRD, INRAE, Aix-en-Provence, 13545, France
[*]A full list of authors appears at the end of the paper.

*Correspondence to*: Felix Martin Hofmann (felix.martin.hofmann@geologie.uni-freiburg.de)

**Abstract.** Beryllium-10 cosmic-ray (CRE) exposure dating has revolutionised our understanding of glacier fluctuations around the globe. A key prerequisite for the successful application of this dating method is the determination of regional production rates of in-situ accumulated $^{10}$Be, usually inferred at independently dated calibration sites. Until now, no calibration site has been available for the mid-elevation mountain ranges of central Europe. We fill this gap by determining in-situ $^{10}$Be concentrations in large boulders on moraines and by applying radiocarbon and infrared stimulated luminescence (IRSL) dating to stratigraphically younger lake sediments in the southern Black Forest, SW Germany. The dating methods yielded concordant results and, based on age-depth modelling with $^{14}$C ages, the age of a cryptotephra, and IRSL ages, we deduced a regional $^{10}$Be production rate in quartz. Calibrating the Black Forest production rate (BFPR) in the cosmic-ray exposure program (CREp) resulted in a spallogenic sea-level and high latitudes (SLHL) production rate of 3.64±0.11 atoms $^{10}$Be g$^{-1}$ quartz a$^{-1}$ when referring to time-dependent Lal/Stone scaling, the European reanalysis (ERA)-40 atmosphere model, and the atmospheric $^{10}$Be-based geomagnetic database in CREp. The BFPR turned out to be ~11% lower than both those at the nearest calibration site in the Alps (4.10±0.10 atoms $^{10}$Be g$^{-1}$ quartz a$^{-1}$ at SLHL) and the canonical global $^{10}$Be production rate (4.11±0.19 atoms $^{10}$Be g$^{-1}$ quartz a$^{-1}$ at SLHL) in CREp. A stronger weathering and snow cover bias and a higher impact of forest, soil, moss, and shrub cover at the study site likely explain this discrepancy.

## 1 Introduction

Beryllium-10 cosmic-ray exposure (CRE) dating is an invaluable tool for age determination in the field of glacial (e.g., Hofmann et al., 2024a), fluvial (e.g., Schoch-Baumann et al., 2022), coastal (e.g., Dawson et al., 2022), and periglacial (e.g., Amschwand et al., 2021) geomorphology as well as for the dating of mass movements (e.g., Hilger et al., 2018). Calculating CRE ages requires effective $^{10}$Be production rates at sampling sites to be determined. CRE age calculators, such as the cosmic-ray exposure program (CREp; Martin et al., 2017) utilise physical models, such as the Lifton-Sato-Dunai (LSD)

scaling scheme (Lifton et al., 2014), to extrapolate $^{10}$Be production rates at calibration sites to sampling sites. The production rate is a key parameter and critically determines the robustness of the ages. At independently dated calibration sites (e.g., Goehring et al., 2012; Claude et al., 2014; Stroeven et al., 2015), geological $^{10}$Be production-rate-calibration allows for the determination of the rate of the in-situ accumulation of cosmogenic $^{10}$Be in rock surfaces with time (cf., Dunai, 2010).

Production-rate-calibration often relies on radiocarbon ages of organic material that was deposited soon after the exposure of the sampled rock surfaces (e.g., Goehring et al., 2012). However, previous studies (e.g., Small and Fabel, 2016a, b; Lowe et al., 2019) have questioned the robustness of radiocarbon ages that have previously been used for geological calibration. Thus, applying a robust approach, ideally involving multiple, independent dating methods, to a potential calibration site is the best practice to overcome dating-method-specific issues. If several lines of evidence converge, the resulting production

rate might be more reliable than a production rate based on a single geochronological method.

Particularly thanks to joint efforts, such as the CRONUS (Cosmic-Ray prOduced NUclide Systematics)-Earth (Phillips et al., 2016) and CRONUS-EU projects (Stuart and Dunai, 2009), the number of $^{10}$Be calibration sites has steadily increased over the last decades (see Martin et al., 2017 for a map of calibration sites). In CREp (Martin et al., 2017), Europe is one of the

regions with the most published regional $^{10}$Be production rates, ranging from 3.48±0.10 (Fenton et al., 2011) to 4.46±0.26 atoms $^{10}$Be g$^{-1}$ quartz a$^{-1}$ at sea-level and high latitudes (SLHL; Borchers et al., 2016) when referring to time-dependent Lal/Stone scaling (Nishiizumi et al., 1989; Lal, 1991; Stone, 2000; Balco et al., 2008), European reanalysis (ERA)-40 atmosphere model (Uppala et al., 2005), and the atmospheric-$^{10}$Be-based geomagnetic database of Muscheler et al. (2005). According to Martin et al. (2017), the majority of the European calibration sites is situated at elevations below 500 metres

above sea-level (m a.s.l.). The only exceptions are Maol Chean Dearg in Scotland (Borchers et al., 2016) and the Chironico landslide in southern Switzerland (Claude et al., 2014), located at elevations of 521 and 761 m a.s.l., respectively. However, for rock surfaces at higher elevation (e.g., Le Roy et al., 2017), snow shielding during winter affects effective production rates more strongly than at lower elevation (cf., Ivy-Ochs et al., 2007). Rates of postdepositional weathering/removal of rock ("erosion" or "rock decay" *sensu* Hall et al. 2012) may also differ from those at production-rate calibration sites. Thus,

extrapolating $^{10}$Be production rates from lower elevation calibration sites to sampling locations at higher elevation may induce an unwanted bias in CRE ages. In addition, soil and vegetation cover (trees, shrubs, and moss) may influence effective production rates. To solve these issues, some studies (e.g., Boxleitner et al., 2019) corrected their ages for these factors, whereas others have presented uncorrected ages and have interpreted them as minimum estimates (e.g., Protin et al., 2019). Other authors performed sensitivity tests to assess the effect of these factors on ages (e.g., Hofmann et al., 2022,

2024b). However, obtaining independent age control is the best strategy to validate proposed correction factors.

The formerly glaciated mid-elevation mountain ranges of Central Europe comprise the Jura and the Variscan mountains, i.e., the Vosges, the Black Forest, the Harz, the Bavarian/Bohemian Forest, and the Giant Mountains (Ehlers et al., 2011). During the Late Pleistocene, the ice masses in this region lay in the mostly non-glaciated corridor between the ice sheets over

northern Europe and the glacier network of the Alps. As previously discussed (e.g., Hofmann et al., 2020, 2022, 2024b; Hofmann, 2023a), there is an urgent need for dating the onset of retreat from their Late Pleistocene maximum positions. Clarifying this issue would help to evaluate the hypothesis that the Alps shielded the ice caps and glaciers from humid air masses from the Mediterranean Sea during the last major advance of piedmont lobe glaciers in the forelands of the Alps (at around 25 ka; e.g., Gaar et al., 2019).

Starting with a pioneering study in the Vosges (Mercier et al., 1999), France, CRE dating became the key method for age determination of ice-marginal positions (e.g., Mercier et al., 2000; Reuther, 2007; Mentlík et al., 2013; see Hofmann et al., 2022 for a compilation), allowing for overcoming three major limitations. First, the lack of organic material prevented radiocarbon dating of moraines. Second, luminescence and radiocarbon dating of deposits present in cirque basins only gave minimum deglaciation ages (e.g., Vočadlová et al., 2015). Third, although investigating the sedimentary sequence in the overdeepened Elbe Valley in the Giant Mountains (Czechia) undoubtedly provided valuable insights into the environmental history, applying luminescence and radiocarbon dating to sediments below and above the till of the last glaciation only resulted in a bracketing age for the last glaciation (cf., Engel et al., 2011). Due to the lack of a regional calibration site, the abovementioned studies have relied on calibration sites in other regions, such as the Chironico landslide in southern part of the Alps (Claude et al., 2014). It should be noted that each calibration site has its own bias with regard to amount and timing of snow cover, the type of vegetation, and rates of weathering. However, the topoclimatic and environmental conditions in the entire Variscan mountains differ from previously established calibration sites (cf., Migoń and Waroszewski, 2022). Introducing a regional calibration site will likely allow for calculating more accurate $^{10}$Be CRE ages for this region.

In this paper, we present a new calibration site in the southern Black Forest, SW Germany (Fig. 1). We chose the Feldsee Cirque (8.0 °E, 47.9 °N WGS 1984 coordinate reference system) because (i) we observed multiple large, quartz-bearing boulders on two well-preserved moraines and because (ii) a bog, the Feldsee Bog, is situated in the tongue basin of the former glacier whose sediments are stratigraphically younger than these ice-marginal moraines (Lang, 2005; Hofmann and Konold, 2023). We first measured the concentration of accumulated, in situ cosmogenic $^{10}$Be in quartz from moraine-boulder surfaces. Obtaining sediment cores from the FSM coring site ("FSM" stands for *Feldseemoor*, the German name of the bog) on a buried moraine, radiocarbon dating of macrofossils, IRSL dating, and establishing an age-depth model with the $^{14}$C ages, the IRSL ages, and the age of a cryptotephra allowed us to derive a minimum age for ice-free conditions at the bog. Based on the $^{10}$Be concentrations in rock samples from the moraine boulders and the modelled basal age of the bog's sediments, we deduced a regional SLHL $^{10}$Be production rate, hereinafter termed the Black Forest production rate (BFPR).

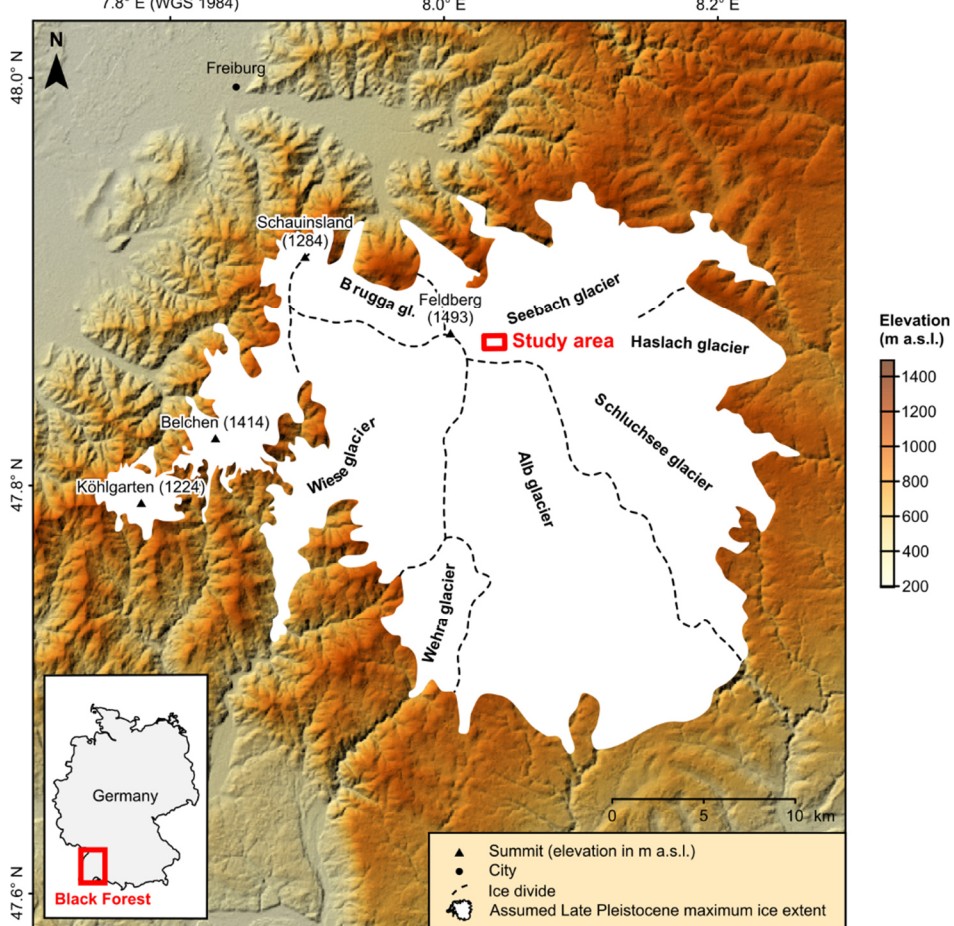

**Figure 1: Topographical map of the southern Black Forest showing the assumed maximum ice extent during the Late Pleistocene (Hemmerle et al., 2016), ice divides (Hemmerle et al., 2016), and outlet glacier names according to the nomenclature of Hofmann et al. (2020). See NASA Jet Propulsion Laboratory (2013) for information on the digital elevation model (DEM) in the background. The inset map shows the location of the Black Forest in Germany.**

## 2 Regional setting

The study site, the Feldsee Cirque, is located in the southern part of the Black Forest in SW Germany (Fig. 1). The Feldsee Cirque is situated about 2 km ESE of Feldberg (1493 m a.s.l.), the highest summit of the Black Forest. Due to the high abundance of glacial landforms (cf., Liehl, 1982; Metz and Saurer, 2012; Hofmann and Konold, 2023), it is a key site for Pleistocene glaciations of the Black Forest. The cirque has attracted glacio-geomorphological and geological research for almost two centuries (Walchner, 1846; Ramsay 1862; Lang et al., 1984; Schreiner, 1990; Hofmann and Konold, 2023).

**2.1 Study site**

The Feldsee Cirque has an impressive, up to 300 m high headwall (Fig. 2; LGRB, 2023). Lake Feldsee, a moraine-dammed
lake up to 33 m deep covers the tongue basin on the cirque floor (Wimmenauer et al., 1990). An about 0.03 km² large bog,
the Feldsee Bog (LUBW, 2006), is located downstream from the ice-marginal moraine at the eastern shore of the lake (Fig.
2). The Feldsee Cirque leads into the Seebach Valley, a well-developed trough valley (Metz, 1985). The study site drains to
the Seebach, originating from above the headwall of the Feldsee Cirque. This stream pertains to the Rhine drainage network
(LUBW, 2022a).


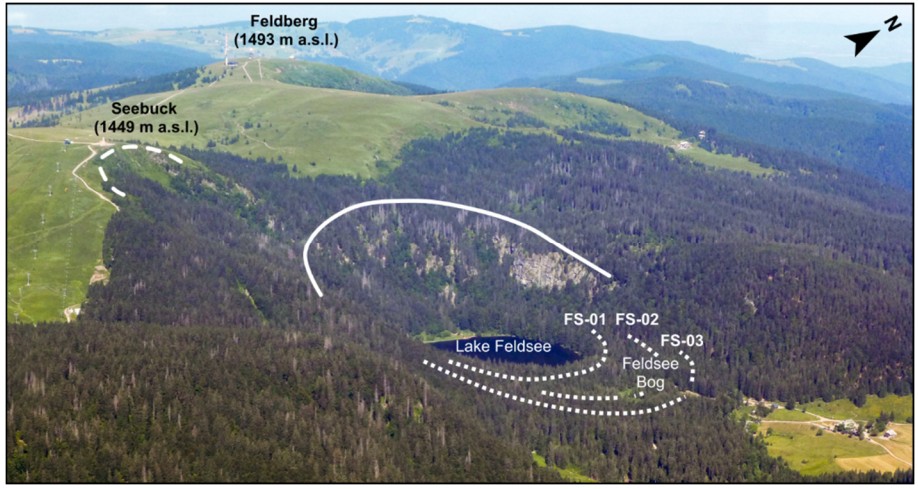

**Figure 2: Oblique aerial photograph of the study site. Moraines are marked with dotted lines. The upper limit of the headwall of
the Feldsee Cirque is marked with a solid line. Lake Feldsee, a moraine-dammed lake, covers the tongue basin on the cirque floor.
Note the prominent moraine at the eastern shore of the lake (position FS-01). The moraine at position FS-02 separates the Feldsee**
**Bog further east into two distinct basins. The bog's sediments have partly buried this moraine. The semi-circular moraine at
position FS-03 bounds the bog to the east. Also note the initial cirque (marked with a dashed line) on the north-eastern flank of
Seebuck. Photo: Matthias Geyer.**

Quartz-bearing rock of the Variscan basement (age: 380—290 Ma; Geyer et al., 2011), i.e., flaser gneiss, migmatite,
porphyry, and paragneiss dominates the study area (LGRB, 2013). In addition, quartz-rich porphyry outcrops on the cirque's
western headwall (LGRB, 2023). Since denudation from about 50 Ma onwards (Eberle et al., 2023) has led to the complete
removal of the Permian, Triassic, and Jurassic sedimentary rock on the Variscan basement (Wimmenauer et al., 1990),
glacial sediments at the study site (mainly till) originate from quartz-rich rock of Variscan age (Schreiner, 1990). During the
Quaternary, the southern Black Forest repeatedly hosted ice caps and glaciers (Metz and Saurer 2012; Hemmerle et al.,
2016; Eberle et al., 2023). For the last (Late Pleistocene) glaciation, previous reconstructions have pointed towards the
temporary existence of four interconnected ice caps (Fig. 1; Liehl, 1982; Metz and Saurer, 2012; Hemmerle et al., 2016),
covering an area of about 1000 km² (Metz and Saurer, 2012). Figure 1 reveals that the ice cap on Feldberg and the

surrounding region was the largest of these ice caps. During the last deglaciation, the ice caps disintegrated into valley glaciers and finally into isolated cirque glaciers (Metz and Saurer, 2012; Hemmerle et al., 2016; Hofmann et al., 2020, 2022, 2024b; Hofmann, 2023a). However, successive phases of ice-marginal stability punctuated the overall trend of glacier recession. According to [10]Be CRE ages for the region NW of Feldberg, formerly covered by the Brugga outlet glacier and the western branch of the Seebach outlet glacier (Fig. 1), repeated phases of glacier recession from moraines occurred no later than 17–16 ka, 14 ka, and by 13 ka at the latest (Hofmann et al., 2022, 2024b; Hofmann, 2023a).

Although the Feldsee cirque glacier at study site area has long since disappeared, snow cover still plays an important role today. Between 1961 and 1990 CE, mean annual precipitation and average annual temperature at the weather station on Feldberg (at 1486 m a.s.l.), situated about 1.5 km west of the study site, amounted to 1909 mm and 3.3 °C, respectively (DWD, 2023). Snowfall accounted for about two thirds of annual precipitation during this period (Matzarakis, 2012). As it will be discussed below, the largely forest-covered Feldsee Cirque thus is a challenging site where seasonal snow cover might have considerably slowed down the accumulation of in situ produced [10]Be in moraine-boulder surfaces.

## 2.2 Previous work

From a glacio-geomorphological perspective, the Feldsee Cirque is undoubtedly one of the most interesting sites in the entire Black Forest. Both Walchner (1846) and Ramsay (1862) already recognised that glaciers shaped the Feldsee Cirque and first described the prominent ice-marginal moraines east of Lake Feldsee (Fig. 3). These landforms document the Feldsee position, the penultimate period of repeated ice-marginal stability during the last deglaciation of the southern Black Forest (Steinmann, 1902; Schrepfer, 1925; Erb, 1948; Liehl, 1982; Metz and Saurer, 2012; Hofmann et al., 2020; Hofmann, 2023b). Hofmann and Konold (2023) recently revisited glacial landforms at the study site and performed geomorphological mapping at the 1:5000 scale with the aid of a DEM derived from light detection and ranging (LiDAR) data (xy-resolution: 1 m; vertical accuracy: ±0.2 m; LGL, 2015) and field evidence. As this is the most recent study on glacial landforms in the study area, we summarise the findings in the following paragraphs (see Fig. 3a).

As only the ice-marginal moraines of the former Feldsee cirque glacier are relevant for the geological calibration of the regional [10]Be production rate, we only discuss ice-marginal positions of this glacier but not those (positions SH-01, SH-02, SH-03, SH-04, and SH-05) of a transfluent glacier that advanced from a south-westerly direction into the Seebach Valley (Fig. 3a; Meinig, 1980).

The ice-marginal moraines of the Feldsee cirque glacier reflect three positions, named, from the oldest to the youngest, FS-03, FS-02, and FS-01 (Fig. 3a; Hofmann and Konold, 2023). A semi-circular and boulder-rich moraine with a sharp crest is located at position FS-03. This moraine partly overlies a moraine at position SH-05. Based on the SW-NE orientation of the latter, this moraine probably formed at the margin of a transfluent ice stream that advanced from the south, i.e., from the

Seehalde Area, from a south-westerly direction into the Seebach Valley. As the moraine at position FS-03 overlies those at position SH-05, the semi-circular moraine at position FS-03 must have formed during a re-advance of the Feldsee cirque glacier. The moraine at position FS-02 subdivides the bog in two parts. Peat and lake sediments partly cover the moraine and, therefore, it is only visible south of the bog, in the centre of the bog (Lang, 2005), and at its north-eastern end. The semi-circular and sharp-crested moraine at position FS-01 dams up Lake Feldsee. According to Schreiner (1990), this landform is one of the morphologically most distinct and best-preserved moraines in the entire southern Black Forest.

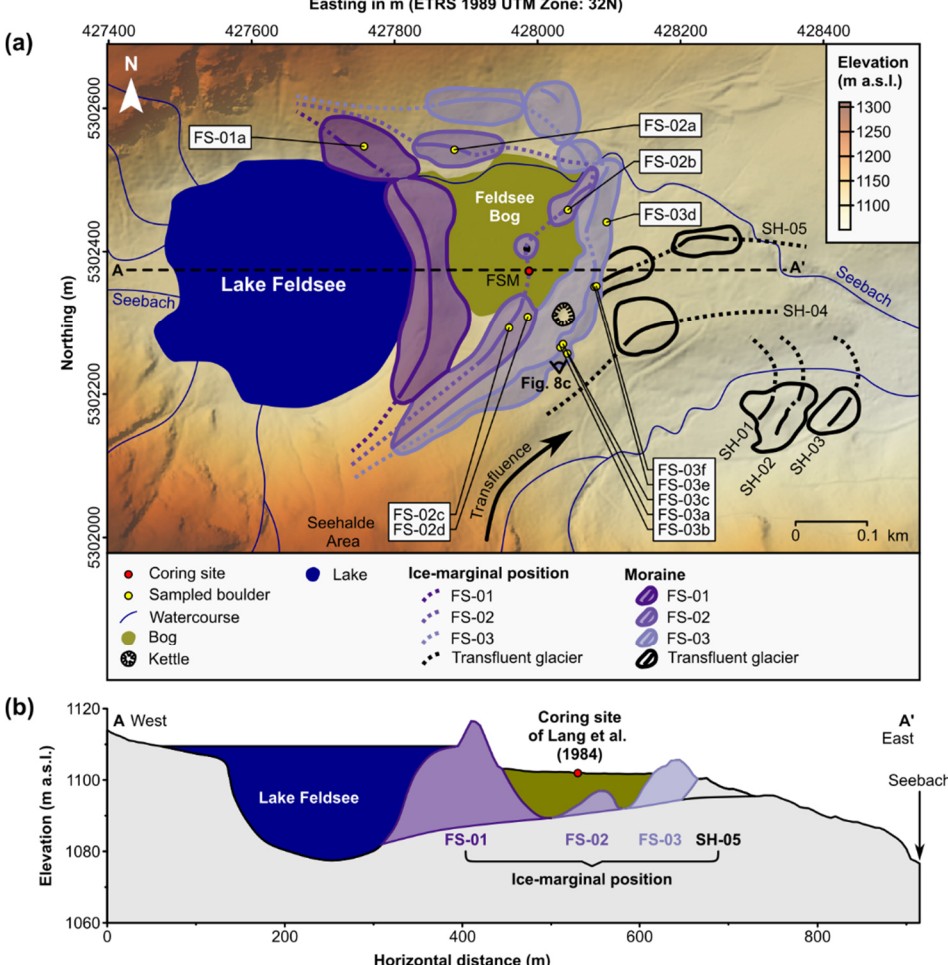

**Figure 3: (a) Topographical map of the study site showing the FSM coring site as well as moraines and ice-marginal positions (Hofmann and Konold, 2023). "FSM" stands for *Feldseemoor*, the German name of the Bog. For the DEM in the background, see LGL (2015). A diagrammatic eye indicates the photo location of Fig. 8c. (a) Transect from the area west of Lake Feldsee to the Seebach (fourfold vertical exaggeration) showing ice-marginal moraines and the coring site for a previous study (Lang, 2005). Note the shallower eastern basin of the Feldsee Bog. See panel (a) for the location.**

In the 1970s and 1980s, Lang et al. (1984) obtained sediment cores at thirteen sites along a W-E transect in the southern portion of the Feldsee Bog and five cores from Lake Feldsee (summarised by Lang, 2005). This research revealed the presence of the buried ice-marginal moraine at position FS-02 (Fig. 3b). Lang (2005) proposed that this landform subdivides the Feldsee Bog into two distinct basins. From the bottom to the top, Lang et al. (1984) identified the following sediments in the cores: till, gyttja, clayey gyttja/clay, gyttja, and peat. At one coring site (coring site "5" in Lang, 2005), these authors

observed a distinct greyish layer at a depth of about 8.1 m below the ground surface (Fig. 4b).

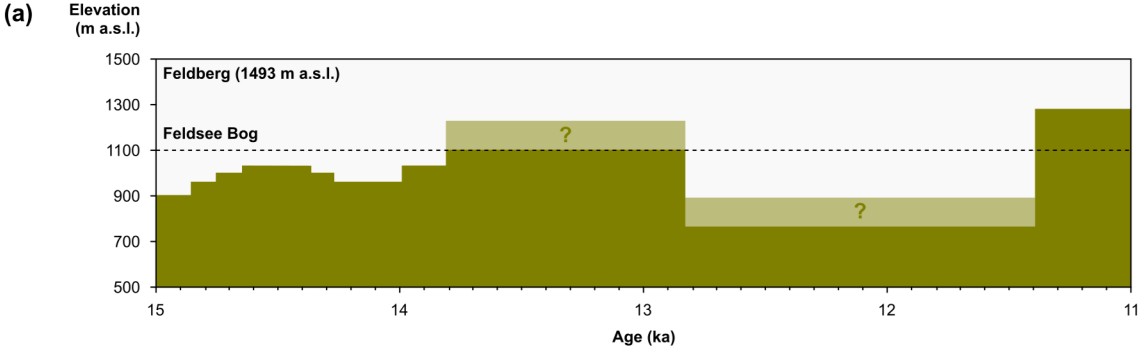

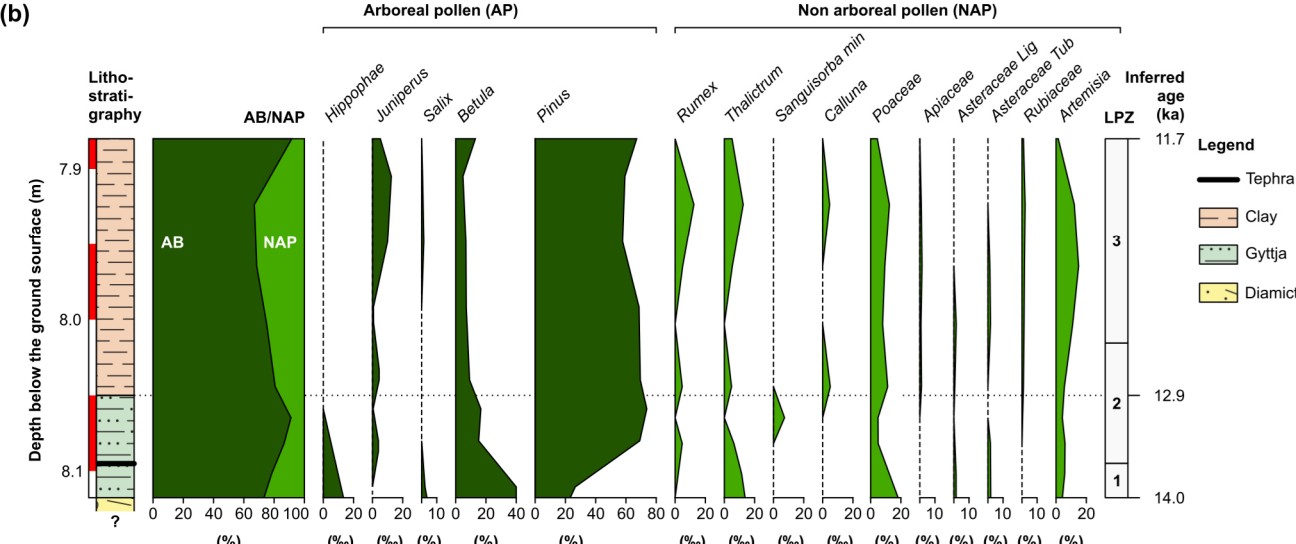

**Figure 4: (a) Evolution of the timberline in the Black Forest between 15 ka and 11 ka (redrawn from Lang, 2006). (b) Lateglacial part of the sediment sequence at a coring site in the Feldsee Bog (Fig. 3b), inferred ages of the sediments, and vegetation dynamics, as documented in the pollen record. LPZ: local pollen zones. Redrawn from Lang (2005).**


Lang et al. (1984) speculated that this layer is the Laacher See Tephra, having a reported age of 13006±9 cal. a BP (Reinig et al., 2021). Unfortunately, these authors did not properly document their observation that would have more convincingly supported the cryptotephra interpretation. Also note that Lang et al. (1984) did not apply radiocarbon dating to their cores.

These authors inferred from the succession of lithostratigraphic units, the position of the apparent tephra and pollen data that a proglacial lake formed between the moraines at positions FS-03 and FS.01 after deglaciation. Sedimentation at the bottom of the lake probably began during the short-lived cooling episode at around 14 ka, often referred to as Older Dryas (cf., Heiri et al., 2014). Later research revealed the presence of a greyish layer in the sediments of Lake Feldsee. Lang (2005) interpreted this layer as the Laacher See Tephra and concluded that both the bog and the lake must have been ice-free no

later than 13.1 ka.

## 3 Methods

To the best knowledge of the authors, the cores obtained by Lang et al. (1984) do, unfortunately, not exist anymore. We thus obtained sediment cores at the FSM coring site during fieldwork in 2021 CE. In addition, we sampled moraine-boulder surfaces and determined the concentration of accumulated, in situ produced $^{10}$Be in quartz.

### 3.1 Coring

Sediment cores at the FSM coring site were obtained with a COBRA vibracorer, allowing for extracting sediment cores with a diameter of 5 cm and a length of 1 m. To prevent daylight exposure of the sediments in the cores, the drilling system was equipped with opaque plastic tubes to store individual sediment cores. The cores were hydraulically extracted, sealed, and opened in the lab with a circular saw in red-light conditions. Sediment samples were obtained from the cores, dried, and loss-

on-ignition (LOI) analyses (cf., Heiri et al., 2001) were undertaken. Weighing the samples prior to drying and before the LOI analyses allowed for the determination of the sediments' water content.

During opening of the cores in the lab, we noted that all sediment cores were shorter than the penetrated depth and, thus, core shortening must have occurred during vibracoring. Generally, core shortening is one of the main limitations of this

technique (cf., Glew et al., 2001). As mentioned by Glew et al (2001), sediments with a higher water content are generally more prone to compaction. We assumed that only the clayey and silty lake sediments in the cores (water content: 18-85%) were affected by shortening and not the stratigraphically older diamicts (water content: 15-17%). Following Glew et al. (2001), we assumed that the sediments in the cores were progressively thinned down-core, i.e., equally affected by compaction. Individual conversion factors were computed for every one-metre-long sediment core which then allowed for

adjusting the thickness of the lithostratigraphic units to the penetrated depth.

### 3.2 Radiocarbon dating

To numerically date the sediments at the coring site, radiocarbon dating of macrofossils was undertaken. Macrofossils were hand-picked after core surface cleaning and KOH pre-treatment of bulk samples. See Figs. S11 to S17 for photos of the macrofossils. A total of nine samples was sent to the radiocarbon laboratory in Poznan, Poland (Table 1) to obtain

accelerator mass spectrometry (AMS) radiocarbon ages. The $^{14}C$ ages were calibrated with the OxCal software (version 4.4; Bronk Ramsey, 2009) available at https://c14.arch.ox.ac.uk/oxcal/OxCal.html (last access: 18 September 2023), and the IntCal20 calibration curve (Reimer et al., 2020).

**Table 1: Macrofossils sampled from the sediment cores from the Feldsee Bog, conventional ages, and calibrated age ranges. Following the recommendation of Millard (2014), 95% ranges of calibration are given.**

| Sample | Decompacted depth (m) | Laboratory code | Material sampled | Conventional age ($^{14}C$ a BP) | | | Calibrated age range (cal. a BP) |
|---|---|---|---|---|---|---|---|
| FSM-450a | 4.28 | Poz-153523 | Unidentified plant fragment | 8620 | ± | 60 | 9750—9490 |
| FSM-450b | 4.28 | Poz-153524 | Unidentified plant fragment | 8650 | ± | 50 | 9750—9530 |
| FSM-536 | 5.24 | Poz-151913 | Unidentified plant fibre | 720 | ± | 30 | - |
| FSM-538 | 5.26 | Poz-152546 | Sphagnum moss | | Modern | | |
| FSM-550 | 5.43 | Poz-152548 | Leaf fragments | 11540 | ± | 120 | 13730—13170 |
| FSM-553 | 5.47 | Poz-152549 | Sphagnum moss | | Modern | | |
| FSM-558 | 5.54 | Poz-152550 | Unidentified plant fragment | 12930 | ± | 70 | 15680—15240 |
| FSM-560 | 5.56 | Poz-152552 | Unidentified plant fragment | 12770 | ± | 60 | 15480—15040 |
| FSM-563 | 5.61 | Poz-152419 | Unidentified plant fragment | 12880 | ± | 60 | 15600—15210 |

### 3.3 IRSL dating

To cross-check the radiocarbon ages, five sediment samples from two of the cores (depth: 4-6 m below the ground surface) were sampled for luminescence dating under subdued red-light, with two further samples taken to account for potential dose-rate inhomogeneity due to the complex stratigraphy (FSM-D1 and FSM-D2; Table 2). The material for equivalent dose ($D_e$) determination was subsequently treated with 10% HCl (no reaction) and 30% $H_2O_2$ (partly heavy reaction) to remove carbonates and organic matter, respectively. For the lowermost two samples, feldspar from the sand fraction (90-200 µm, CG) and polymineral fine-grains (4-11 µm, FG; based on grain-size distributions) were extracted. For the remaining three samples, only fine-grains were extracted. Measurements were done on a Freiberg Instruments Lexsyg Research (Richter et al., 2013), equipped with a ET935QB photomultiplier (Hamamatsu Photonics) and using the combination of a Schott BG39 (3 mm) and a 414/46 BrightLine HC interference filter (3.5 mm) for detection.

Table 2: Samples for IRSL dating. Depth refers to the depth below surface after decompaction. Stim.: stimulation used for Equivalent Dose ($D_e$) determination. Given are the measured water content (W) and the sediment moisture (Mois.) used for dose rate calculations. The activity of different isotopes (K, Th, U-238, Ra-226) is given, revealing disequilibrium in the Uranium decay chain. *n*: number of replicate $D_e$ measurements, Model: applied approach to extract mean $D_e$ (CAM: Central Age Model; MAM: Minimum Age Model).

| Lab code | Depth (m) | Stim. | W (%) | Mois. (%) | K (Bq kg⁻¹) | Th (Bq kg⁻¹) | U-238 (Bq kg⁻¹) | Ra-226 (Bq kg⁻¹) | Dose rate (Gy ka⁻¹) | n | OD (%) | Model | De (Gy) | Age (ka) |
|---|---|---|---|---|---|---|---|---|---|---|---|---|---|---|
| FSM-D1 | 4.73-4.78 | - | 80 | 100±10 | 230±40 | 51±5 | 490±40 | 59±5 | | - | - | - | - | - |
| FSM1-FG | 4.83-4.88 | IRSL | 42 | 60±5 | 880±100 | 81±6 | 174±19 | 92±7 | 4.55±0.37 | 5 | 1 | CAM | 36.67±0.55 | 12.2±1.0 |
| | | IR-50 | | | | | | | | 4 | 0 | CAM | 37.17±0.49 | 11.5±1.0 |
| | | pIR | | | | | | | 4.61±0.38 | 4 | 0 | CAM | 55.19±0.75 | 11.9±1.0 |
| FSM2-FG | 4.88-4.93 | IRSL | 46 | 60±5 | 640±80 | 85±7 | 201±22 | 100±5 | 4.77±0.40 | 5 | 2 | CAM | 35.33±0.59 | 10.4±0.9 |
| | | IR-50 | | | | | | | | 5 | 2 | CAM | 36.42±0.54 | 10.7±0.9 |
| | | pIR | | | | | | | | 5 | 0 | CAM | 56.77±0.69 | 11.8±1.0 |
| FSM-D2 | 5.23-5.28 | - | 76 | 100±10 | 310±50 | 57±5 | 260±30 | 73±7 | | - | - | - | - | - |
| FSM3-FG | 5.33-5.38 | IRSL | 64 | 80±5 | 490±60 | 66±5 | 282±26 | 98±7 | 3.74±0.34 | 5 | 8 | CAM | 27.33±1.04 | 10.2±0.7 |
| | | IR-50 | | | | | | | | 5 | 0 | CAM | 31.48±0.58 | 11.7±1.0 |
| | | pIR | | | | | | | | 5 | 2 | CAM | 55.38±0.89 | 14.1±1.2 |
| FSM4-FG | 5.43-5.48 | IRSL | 51 | 70±5 | 640±70 | 67±5 | 215±20 | 84±6 | 4.02±0.32 | 5 | 3 | CAM | 36.49±0.75 | 12.7±1.0 |
| | | IR-50 | | | | | | | | 5 | 0 | CAM | 38.42±0.76 | 13.2±1.0 |
| | | pIR | | | | | | | | 5 | 2 | CAM | 61.17±1.28 | 15.3±1.2 |
| FSM4-CG | | IRSL | | | | | | | 3.45±0.22 | 20 | 32 | MAM | 30.99±1.57 | 12.7±0.9 |
| FSM5-FG | 5.53-5.58 | IRSL | 46 | 55±5 | 680±70 | 67±5 | 233±20 | 80±5 | 5.65±0.46 | 5 | 0 | CAM | 45.33±0.60 | 14.0±1.2 |
| | | IR-50 | | | | | | | | 4 | 2 | CAM | 46.41±0.76 | 14.2±1.2 |
| | | pIR | | | | | | | | 4 | 0 | CAM | 78.12±1.08 | 17.6±1.3 |
| FSM5-CG | | IRSL | | | | | | | 3.96±0.41 | 20 | 24 | MAM | 44.28±4.28 | 15.8±1.7 |

It should be noted that optically stimulated luminescence measurements on quartz revealed no suitable signal, similar to reports on other directly bedrock derived samples (e.g., Preusser et al., 2006) and experience from the nearby Upper Rhine Graben (Preusser et al., 2016, 2021). Therefore, feldspar was selected as dosimeter. For all samples, a standard IRSL

protocol (modified from Preusser, 2003) was applied. This protocol comprised a preheat to 250 °C for 60 s and IRSL stimulation at 50 °C for 100 s (IR-50). For fine grains, a post-IR (pIR) IRSL protocol was additionally tested to potentially overcome the need for fading correction. This protocol involved a preheat to 250 °C for 60 s, IRSL stimulation at 50 °C for 100 s, and a second stimulation at 225 °C for 100 s (pIR). For fine grains, the five replicate measurements were considered sufficient due to the excellent reproducibility, whereas for the sand fraction 20 replicate measurements were performed. For most samples, the average dose was calculated by the Central Age Model (CAM). In two cases, where the overdispersion exceeded 20%, the Minimum Age Model (MAN) was allied with a *sigma_b* value of 0.20 (Galbraith and Roberts, 2012). For IRSL and IR-50, a fading correction was undertaken using a g-value of 3.7±0.2 g per decade.

The concentration of dose-rate relevant elements was determined using a high-resolution gamma spectrometer, as outlined in Preusser et al. (2023). An alpha efficiency of 0.07±0.02 and an internal K-content of 12.5±1.0% (Huntley and Baril, 1997) were assumed. Note that the setting is particularly challenging with regard to dose rate determination. While average sediment moisture during burial time was estimated based on the water content measured directly after opening the cores in the laboratory, it should be considered that a certain loess of water may have occurred during and after the coring operations. Furthermore, the deposits will have compacted after initial deposition due to the increase of load with time. Hence, the water used in the calculations are higher that the measured but still only represent an approximation. All age calculations were performed with the ADELEv2017 software (Degering and Degering, 2020). As the different sediment layers are partly quite thin, three-layer models were used for all samples, as implemented in ADELEv2017. Namely, for each layer an individual dose rate was calculated from which the effective dose rate acting on the sample area was determined.

In addition, evidence for significant radioactive disequilibrium in the Uranium decay chain was observed for most of the samples. This is common in organic-rich deposits (e.g., Preusser and Degering, 2007; Preusser et al., 2023) and reflects the absorption of Uranium by humic substances from water (cf., Ivanovich and Harmon, 1992). This was accounted for in the dose rate calculations for each individual layer assuming a constant uptake of Uranium since the time of deposition. Both the applied layer model and correction of radioactive disequilibrium represent only approximations to the real situation, as it is unfeasible to determine the detailed 3D geometry of the deposits, and the exact timing of Uranium uptake.

### 3.4 Age-depth modelling

To obtain a minimum age for the moraine at position FS-02, age-depth modelling with the online Oxcal programme and IntCal20 (Reimer et al., 2020) was undertaken by interrogating the [14]C ages of macrofossils, the age of the Laacher See Tephra (13006±9 cal. a BP; Reinig et al., 2021), and the IRSL ages (converted in cal. a BP). Two IRSL ages were available for the FSM4 and FSM5 samples, i.e., ages for both coarse-grains and fine-grains. Error-weighted IRSL ages and uncertainties of the error-weighted IRSL ages were computed prior to age-depth modelling. The *P_Sequence* function was selected in Oxcal, as the [14]C ages of the macrofossils and the IRSL ages of sediment samples were expected to increase with

depth. In addition, this function allows for changes in the sedimentation rate (Bronk Ramsey, 2008). Following the guidelines of Bronk Ramsey and Lee (2013), the model was tuned to find the most suitable value for the *k*-factor, i.e., the number of increments per depth unit. See the supplement for the input-code for Oxcal.

### 3.5 Regional [10]Be production rate calibration and CRE dating of moraines

For establishing the BFPR, we collected surface-rock samples (Table 3) from (i) six gneiss boulders on the moraine at position FS-03 and (ii) four gneiss boulders on the ice-marginal moraine at position FS-02. We also sampled the surface of the FS-01a gneiss boulder on the moraine at position FS-01a for age calculations. However, we did not include the sample in the calibration dataset, as moraine formation might have post-dated the onset of deposition of lake sediments at the FSM coring site (Fig. 3b). The modelled basal age of the Feldsee Bog then offered the opportunity to derive a regional SLHL spallogenic [10]Be production rate.

#### 3.5.1 Fieldwork, sample preparation, and measurements

Since the study of Tomkins et al. (2021) demonstrated that landform stability mainly influences the scatter in age distributions from moraines (and thus in [10]Be concentrations), only well-embedded boulders were selected to avoid underestimated [10]Be concentrations due to boulder rotating as well as post-depositional and post-stabilisation exhumation. As the moraine at position FS-03 consisted of clast-supported diamicts and some parts of the moraine were solely composed of boulders, identifying stable and large boulders proved to be straightforward. The same was true for the moraine at position FS-02 although this landform consisted of matrix-supported diamicts. Identifying large and stable boulders on the moraine at position FS-01 turned out to be difficult, as this landform consisted of matrix-rich diamicts and exhibited only a few boulders. We thus only sampled one large boulder. Hofmann and Konold (2023) mapped a kettle on the proximal side of the moraine at position FS-03 and on the moraine at position FS-02 in the centre of the Feldsee Bog (Fig. 3), pointing to paraglacial reworking and delayed moraine stabilisation (cf., Porter et al., 2019). To minimise the risk for paraglacial reworking issues, we avoided sampling boulders in the vicinity of these landforms.

Previous sampling guidelines suggested that flat-topped boulders should be selected for CRE dating of moraines (e.g., Ivy-Ochs and Kober, 2008). However, none of the moraine boulders in the study area was flat-topped, thus rock samples (Table 4) were obtained with an angle grinder, a chisel, and a hammer from dipping surfaces with a constant angle. See the detailed sample documentation (Tables S1 to S11) for strike and dip of the sampling surfaces, measured with a geological compass. Masarik and Wieler (2003) argued that the cosmic ray flux on non-flat rock surfaces is lower than on flat rock surfaces, since the number of scattered neutrons is larger in non-flat rock surfaces. Due to these potential edge effects, rock surfaces near boulder edges were avoided during fieldwork. As scaling of the [10]Be production rate to sampling sites required coordinates of the sampling sites, *xy*-coordinates of the boulders were recorded with a global navigation satellite system (Leica CS20 controller and Leica Viba GS14 antenna).

**Table 3: Location of the sampled moraine boulders and their height, thickness of the rock samples, and topographic shielding at the sampling sites.**

| Ice-marginal position | Boulder | xy-coordinate (WGS 1984 coordinate reference system) | | Elevation (m a.s.l.) | Boulder height (m) | Sample thickness (cm) | Topographic shielding factor |
| --- | --- | --- | --- | --- | --- | --- | --- |
| | | Latitude (° N) | Longitude (° E) | | | | |
| FS-01 | FS-01a | 47.872193 | 8.033919 | 1128 | 1.6 | 1.5 | 0.958616 |
| FS-02 | FS-02a | 47.872164 | 8.035611 | 1113 | 1.1 | 2.4 | 0.947912 |
| | FS-02b | 47.871428 | 8.037746 | 1103 | 0.9 | 2.6 | 0.987496 |
| | FS-02c | 47.869931 | 8.036674 | 1108 | 1.4 | 2.0 | 0.987583 |
| | FS-02d | 47.870075 | 8.037022 | 1107 | 1.8 | 1.9 | 0.988521 |
| FS-03 | FS-03a | 47.869613 | 8.037770 | 1108 | 2.2 | 2.1 | 0.989194 |
| | FS-03b | 47.869690 | 8.037641 | 1112 | 2.5 | 2.1 | 0.954505 |
| | FS-03c | 47.869732 | 8.037687 | 1110 | 3.9 | 1.9 | 0.985596 |
| | FS-03d | 47.871280 | 8.038474 | 1099 | 0.9 | 2.0 | 0.989967 |
| | FS-03e | 47.870467 | 8.038263 | 1106 | 2.0 | 2.4 | 0.960224 |
| | FS-03f | 47.870475 | 8.038298 | 1106 | 2.2 | 2.8 | 0.985626 |

320

The elevation of the sampling sites was later retrieved from the abovementioned high-resolution digital elevation model of the study site of the Baden–Württemberg State Agency for Spatial Information and Rural Development (LGL, 2015). Obtaining pairs of azimuth and elevation angles of the horizon around the sampling sites for topographic shielding factor calculations was not possible at all sampling sites due to the presence of dense mixed forests. Therefore, the ArcGIS toolbox

325 of Li (2018) was chosen for shielding factor calculations, considering (i) self-shielding of dipping surfaces and (ii) shielding by topographical obstructions around the sampling sites. Following the guidelines of Hofmann (2022), the high-resolution DEM of the study site (LGL, 2015) was resampled to a xy-resolution of 30 m for shielding factor calculations. The mass and thickness of each rock fragment in the samples was determined to compute the mass-weighted average of the sample's thickness. Since the sampled boulders were rich in quartz, the density of quartz (2.65 g cm$^{-3}$) was assumed for the thickness

330 correction. For photos of the boulders, dimensions of the boulders, and height above ground of the sampling surfaces, see the detailed sample documentation in the supplement.

Quartz separation and $^{10}$Be extraction from purified quartz was accomplished in the laboratory facilities of the University of Freiburg (Germany) and the Laboratoire National des Nucléides Cosmogéniques (LN$_2$C) in Aix-en-Provence (France) according to the protocol described in Hofmann et al. (2024b). After crushing, and wet sieving (target grain size: 0.25-1 mm), the samples were passed through a magnetic separator (S.G. Frantz Co.). The samples were then treated with mixtures of 37% HCl and 35% H$_2$SiF$_6$ to further isolate quartz and remove feldspars. Because the samples still contained feldspar after this treatment, the samples were subsequently etched with diluted 5.5% HF, dried, spiked with magnetite powder (325 mesh), and passed through a magnetic separator. Meteoric $^{10}$Be was removed with 48% HF in three steps, with 10% of the quartz dissolved in each step. A dose of about 150 mg of a $^9$Be carrier solution (3025±9 µg $^9$Be g$^{-1}$) was added to each sample before total dissolution with 48% HF. Chromatography with anionic and cationic exchange resins (DOWEX 1X8 and 50WX8), as well as precipitation stages were then performed to further separate and purify beryllium. To thermally decompose the final Be(OH)$_2$ precipitate to BeO, the samples were subsequently heated to 700 °C. The samples were finally mixed with Nb and pressed into copper cathodes for AMS measurements.

The $^{10}$Be concentrations in the samples (Table 4) were deduced from AMS measurements at ASTER (Accélérateur pour les Sciences de la Terre, Environnement, Risques; Arnold et al., 2013), the French AMS national facility, at CEREGE (Centre Européen de Recherche et d'Enseignement des Geosciences de l'Environnement) in Aix-en-Provence (France). The measured $^{10}$Be/$^9$Be ratios were normalised with respect to the in-house standard STD-11 using an assigned $^{10}$Be/$^9$Be ratio of (1.191±0.013) × 10$^{-11}$ (Braucher et al., 2015) and the Be half-life of (1.387±0.012) × 10$^6$ years (Chmeleff et al., 2010; Korschinek et al., 2010). The $^{10}$Be/$^9$Be ratio uncertainties comprise (i) the measurement uncertainty (counting statistics), (ii) the error of average standard measures, and (iii) the systematic error of ASTER (0.5%; Arnold et al., 2010). The $^{10}$Be concentrations in the samples were corrected for the $^{10}$Be concentration in a batch-specific chemical blank (Table 4).

### 3.5.2 Production rate calibration

The calibration of the spallogenic $^{10}$Be SLHL Black Forest production rate followed the workflow of Martin et al. (2017, their Fig. 3). Following Eq. 4 in Martin et al. (2017), the $^{10}$Be concentration in each sample was first corrected for the sample thickness and for topographic shielding to compute a theoretical $^{10}$Be concentration in a sample with null thickness. The $^{10}$Be concentrations were then scaled to the mean latitude, longitude, and elevation representative for all sampled boulders (Martin et al., 2017). Following the guidelines of Ross (2003), the $^{10}$Be concentrations were subsequently evaluated with Peirce's criterion (Peirce, 1852), and a weighted $^{10}$Be concentration was computed after the exclusion of outliers. The mean squared weighted deviation (MSWD) for the $^{10}$Be concentrations turned out to be higher than one. As recommended by Martin et al. (2017), the standard error of the weighted mean $^{10}$Be concentration (calculated with 1σ uncertainties of the $^{10}$Be concentrations) was multiplied with $\sqrt{MSWD}$ to obtain the uncertainty of the average $^{10}$Be concentration.

**Table 4: Characteristics of the samples, $^9$Be added, the results of AMS measurements, blank-corrected $^{10}$Be concentrations, and scaling factors for the samples in the calibration dataset (geomagnetic database: Muscheler et al., 2005). $^{10}$Be concentrations in the samples were corrected with the $^{10}$Be concentration in a batch-specific blank. \*Analytical blank. \*\* The $^{10}$Be concentration was classified as an outlier with Peirce's criterion (Peirce, 1852). \*\*\*Due to low $^9$Be currents during AMS measurements, the reported $^{10}$Be concentration should not be regarded as reliable. \*\*\*\*This concentration was not used for production rate calibration.**

| Ice-marginal position | Sample | Mass of dissolved quartz (g) | Beryllium-9 added (× 10$^{19}$ atoms) | Beryllium-10/ Beryllium-9 ratio (× 10$^{-14}$) | Blank-corrected $^{10}$Be concentration (atoms g$^{-1}$ quartz) | Normalised blank-corrected $^{10}$Be concentration (atoms g$^{-1}$ quartz at SLHL) | Scaling factor for the whole duration of exposure | |
|---|---|---|---|---|---|---|---|---|
| | | | | | | | Time-dependent Lal/Stone scaling (Nishiizumi et al., 1989; Lal, 1991; Stone, 2000; Balco et al., 2008) | LSD scaling (Lifton et al., 2014) |
| - | BK-FS-SH-WH-WK* | - | 3.03 | 0.139 ± 0.032 | - | - | - | |
| FS-01 | FS-01a | 12.0489 | 3.00 | 5.54 ± 0.18 | 134500 ± 4700 | - \*\*\*\* | - | |
| FS-02 | FS-02a | 9.9458 | 3.03 | 4.60 ± 0.17 | 135800 ± 5300 | 56600 ± 5300 | 2.58 | 2.59 |
| | FS-02b | 26.1755 | 3.05 | 12.21 ± 0.45 | 140600 ± 5300 | 59100 ± 5300 | 2.56 | 2.56 |
| | FS-02c | 21.7892 | 3.03 | 10.37 ± 0.32 | 142100 ± 4600 | 56900 ± 4600 | 2.57 | 2.58 |
| | FS-02d | 16.0494 | 3.02 | 5.05 ± 0.22 | 92300 ± 4200 \*\* | 36800 ± 4200 \*\* | 2.58 | 2.57 |
| FS-03 | FS-03a | 2.0943 | 3.03 | 1.25 ± 0.08 | 161000 ± 13000 | 64000 ± 13000 | 2.58 | 2.58 |
| | FS-03b | 13.1496 | 3.03 | 5.26 ± 0.70 | 118000 ± 16000 \*\*\* | - \*\*\* | - | |
| | FS-03c | 21.2975 | 3.03 | 10.10 ± 0.31 | 141500 ± 4600 | 56200 ± 4600 | 2.59 | 2.58 |
| | FS-03d | 16.2466 | 3.02 | 7.32 ± 0.24 | 133300 ± 4700 | 53400 ± 4700 | 2.56 | 2.56 |
| | FS-03e | 10.2760 | 3.02 | 5.05 ± 0.18 | 144100 ± 5400 | 59400 ± 5400 | 2.58 | 2.57 |
| | FS-03f | 9.1148 | 3.01 | 4.60 ± 0.19 | 147400 ± 6400 | 59400 ± 6400 | 2.58 | 2.57 |

The regional $^{10}$Be production rate was calculated with the aid of CREp, available at https://crep.otelo.univ-lorraine.fr (last access: 6 November 2023). For calibration, the modelled basal age (15640±420 cal. a BP) of the lake sediments above the moraine at ice-marginal position FS-02 was converted into ka before 2010 CE. The spallogenic SLHL $^{10}$Be BFPR in quartz

was computed for the scaling schemes in CREp, i.e., time-dependent Lal/Stone (Nishiizumi et al., 1989; Lal, 1991; Stone, 2000; Balco et al., 2008) and LSD (Lifton et al., 2014) scaling, and all geomagnetic databases in CREp, i.e., the atmospheric [10]Be-based virtual dipole moment (VDP; Muscheler et al., 2005, and references therein), the LSD framework (Lifton et al., 2014), and the Lifton 2016 VDM (Lifton, 2016, and references therein). Following the approach in a previous calibration study (Fenton et al., 2011), a "baseline" production rate was first calculated, i.e., a production rate that accounts for the site-specific bias induced by snow cover, vegetation cover, soil cover, and postdepositional weathering and by other factors, such as changes in atmospheric circulation.

As many geomorphologists prefer to work with version 3 of the online exposure age calculator, formerly known as the CRONUS-Earth online exposure age calculator (available at http://hess.ess.washington.edu/math/v3/v3_cal_in.html, last access: 8 November 2023), the "baseline" production rate was additionally determined with this calculator. See the supplement for the input-sheet.

Fenton et al. (2011) argued that snow cover and post-depositional weathering of sampling surfaces probably affected their regional spallogenic SLHL [10]Be production rates at two rock avalanches in Norway, and proposed postdepositional-weathering- and snow-cover-corrected regional spallogenic SLHL production rates. The sampling sites in the southern Black Forest were situated in a heavily forested and sheltered area. A few centimetre-thick organic soil, mosses, and shrubs, most notably blueberry bushes (*Vaccinium myrtillus)* covered the sampled boulders. As soil cover on boulders leads to enhanced chemical weathering rates (cf., Ahnert, 2009), the boulders probably underwent significant postdepositional weathering. In addition, snow cover during winter has probably slowed down the accumulation of [10]Be in quartz in the sampled boulders. During field surveys in 2022 CE, we observed a substantial snow cover in winter on the boulders, being up to a few decimetres thick.

Therefore, proposing a postdepositional-weathering- and snow-cover-corrected regional [10]Be production rate was deemed mandatory. On the FS-02b boulder, we observed a protruding quartz vein with a height of 1 cm. The height of the quartz vein and the abovementioned modelled basal age of the lake sediments at the FSM coring implied a post-depositional weathering rate of 0.06 cm ka[-1] for the weathering-corrected BFPR. It should be noted that this rate of weathering and removal of rock is lower than a previous rate for a similar setting in the Bavarian Forest (0.24 cm ka[-1]; recalculated from Reuther, 2007). According to data from the weather station of the DWD in the municipality of Schluchsee (at 990 m a.s.l.), situated about 10 km to the SE of the study area, seasonal cover lasted for four months in the 1961-1990 CE period (average snow depth: 0.3 m; DWD, 2023). Assuming similar conditions during the whole duration of exposure, a snow density of 0.3 g cm[-3], an attenuation length for fast neutrons in snow of 109 g cm[-2] (Zweck et al., 2013), the commonly used equation 3.76 in Gosse and Phillips (2001) yielded a snow shielding factor of ~0.974. This factor was adopted for production rate calibration.

Modelling has revealed up to a 7% reduction in the intensity in cosmic ray flux in old-growth temperate forests (Plug et al., 2007). As the sampled moraine boulders were situated in forested areas, a vegetation-corrected production rate was calculated. According to palynological data from several bogs and fens in the Black Forest, the timberline rose to an elevation of ≥1100 m a.s.l. during the last interstadial of the Late Pleistocene (Fig. 4a, Lang, 2006), commonly referred to as Allerød interstadial (13.9—12.8 ka; Heiri et al., 2014). During this period, open pine-birch forests probably covered the study site (Fig. 4b). The presence of *Pinus* in the pollen record, however, does not necessarily indicate that the study site was forested. As discussed by Andrieu et al. (1997), *Pinus* generally tends to be overrepresented in pollen records due to high pollen production and winds easily transport the pollen over long distances. Unfortunately, Lang (2005) did not observe *Pinus* or *Betula* macro-remains that would have more convincingly supported the idea of open pine-birch forests. Lang (2006) suggested that the timberline shifted back to about 750 m a.s.l. during the subsequent cold phase (Fig. 4a), often referred to as Younger Dryas stadial (12.8—11.7 ka; Heiri et al., 2014). However, with regard to the Northern Alps where the timberline lowered by about 200 m during this cold phase (cf., Stojakowits et al., 2014), the reconstructed shift of the timberline in the Black Forest (c. 350 m) appears quite strong. It is therefore possible that the study site remained a sparsely forested area (Stojakowits, P., pers. comm., 25 October 2023). Since we could not conclusively clarify the Lateglacial vegetation history of the study site, we assumed for the vegetation correction that forests covered the study area throughout the duration of exposure. We are, however, aware that this correction might overestimate the impact of the vegetation on the cosmic ray flux at the sampling sites. For the vegetation correction, we assumed a shielding factor of 0.98 for boreal forests (Plug et al., 2007), the probably best analogues for the forests at the study site.

### 3.6 Assessment of the impact of the new production rate

To assess the impact of the newly calibrated BFPR on CRE ages, CRE ages, internal (analytical) uncertainties, and external uncertainties (i.e., analytical uncertainties plus the error of the $^{10}$Be production rate added in quadrature) for the sampled moraine-boulder surfaces were calculated with the Chironico landslide spallogenic SLHL production rate and the BFPR. The following parameters were chosen to scale the SLHL production rates to the sampling sites: time-dependent 'Lm' scaling (Nishiizumi et al., 1989; Lal, 1991; Stone, 2000; Balco et al., 2008) and the ERA atmosphere model (Uppala et al., 2005), as recommended by Martin et al. (2017). The ages were corrected for changes in the Earth's magnetic field with data from the atmospheric $^{10}$Be-based geomagnetic database (Muscheler et al., 2005). Since the sampled moraine boulders were derived from quartz-rich lithologies, the density of quartz (2.65 g cm$^{-3}$) was assumed for the thickness correction.

### 3.7 Statistical assessment of age datasets

Multiple ages were available for most of the moraines. The assessment of the ages followed the guidelines of Balco (2023). If three or more ages were available from the same landform, reduced chi-squared ($\chi^2_R$) was computed. $X_R^2$ was then

compared with a critical value from a standard $\chi^2$-table (degree of freedom: $n - 1$). The confidence interval was set to 95%. If $\chi^2{}_R$ turned out to be lower than the critical value, the hypothesis that the data formed a single population was at 95% confidence (cf., Balco, 2011). In this case, the landform age was determined by computing the error-weighted mean. If $\chi^2{}_R$ turned out to be higher than the critical value, implying that measurement uncertainties did not fully account for scatter in ages from the same landform (cf., Balco, 2011), the age that was furthest from the average age in relation to its measurement uncertainty was considered an outlier and removed from the dataset. This procedure was repeated until the pruned dataset yielded an acceptable $\chi^2{}_R$ value. Not more than half of the ages from the same landform were excluded. The procedure was also stopped if three ages were remaining. If the original or pruned datasets yielded a $p$-value $> 0.05$, the internal landform uncertainty was determined by calculating the standard error. If the original or pruned datasets gave a $p$-value $< 0.05$, the standard deviation of the ages was selected as internal landform age uncertainty. The external landform age uncertainties were calculated by adding the internal landform age uncertainties and the error of the selected production rate in quadrature.

## 4 Results

Vibracoring at the FSM coring site allowed for obtaining sediment cores with a total length of 8 m. Borehole FSM recovered the sedimentary succession of the Feldsee Bog and the uppermost 0.32 m of the partly buried moraine at position FS-02 (Fig. 5). The percentage of sediment recovery increased from 67% (depth: 4-5 m) to 81% (depth: 5-6 m). Decompaction was thus undertaken with correction factors of 1.49 (4-5 m) and 1.39 (5.00-5.68 m). Beryllium-10 concentrations in a total of ten moraine-boulder surfaces were successfully determined, allowing for production rate calibration.

### 4.1 Lithostratigraphy of the sediment sequence

In the lowermost 0.32 m, this succession consisted of greenish grey, medium-densely packed silty diamicts that were mostly clast-supported and occurred in a fining-upwards package. Individual clasts were angular to subangular, and of diverse crystalline lithologies of local origin (gneiss, migmatite and granite porphyry; LGRB, 2013). With 1.2% to 1.9%, LOI values in samples from the sandy diamicts at a depth of 5.80–5.68 m were very low. At a depth of 5.68 m, the diamicts transitioned into crudely laminated light grey to olive grey fines, silt, and clay, that contained dispersed coarse sand to fine gravel clasts at their base. LOI values varied between 1.7% and 10.5 %. Dark brown to grey fines, silt, and clay overlay these deposits. LOI values ranged from 12.3% to 35.7%. Macroscopic plant fragments and a distinct 2-3 mm-thick light-grey lamina occurred in this unit. Tephra observed in non-visible ash beds (cryptotephra; cf., Krüger and van den Bogaard, 2021) at a decompacted depth of 4.91 m were glassy, generally <100μm in size, and rich in fluid inclusions and vesicles (Fig. 5). They were isotropic and often conchoidally fractured. Oftentimes the tephra shards featured elongated fluid inclusions that were directionally aligned. Light beige grey fines (sand, silt, and clay) with low LOI values (4.9–5.6 %) overlay the dark brown to grey fines. The basal contact of this unit was sharp. Starting at a depth of 4.75 m, organic-rich (LOI: 30.5–54.8%) gyttja-type deposits occurred that were crudely bedded on a cm-scale (Fig. 5).

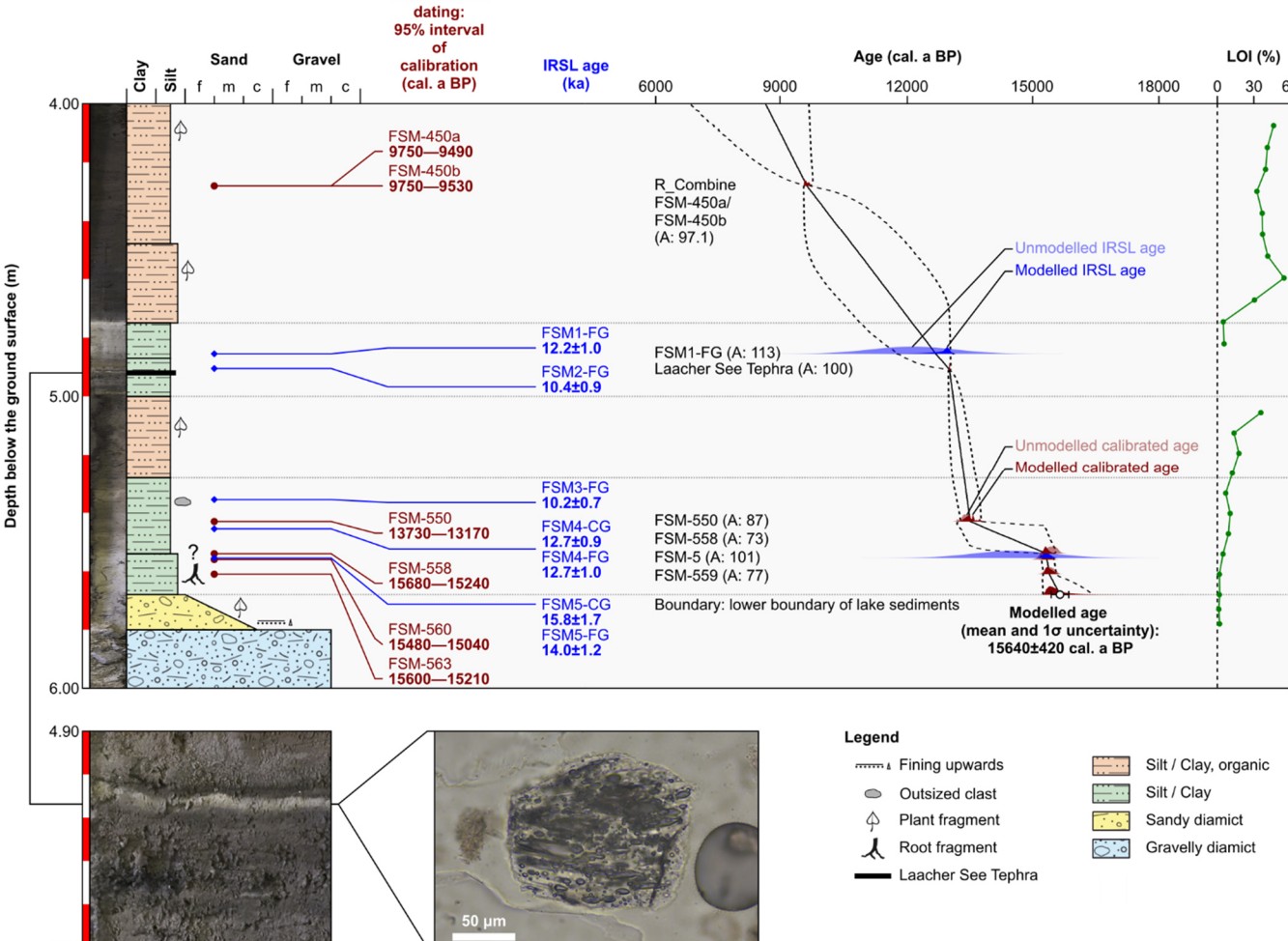

**Figure 5: Sediment sequence at the FSM coring site, calibrated ages (95% ranges of calibration), IRSL ages, the age-depth model, and LOI. Photos of the sediment cores from the Feldsee Bog were acquired with the methodology of Gegg and Gegg (2023). 95% ranges of the modelled ages are marked with dashed lines. The solid line represents the mean modelled age. Agreement indices (A) for individual ages are given in parentheses.**

The succession of FSM (Fig. 5) reflects the glacial-postglacial transition of the study area. The FSM borehole was situated in elongation of a geomorphologically distinct ice-marginal moraine (Fig. 3a) and recovered its continuation into the Feldsee Bog. The corresponding diamicts (6.00-5.68 m) were medium-densely packed and contained washed-out sections with decreased fines content. This indicates an origin from melt-out and subaqueous mass movements (e.g., Schlüchter, 1997). The decreasing gravel content in this cycle further suggests a progressive shift from an immediately ice-proximal to a slightly more distal position due to glacier retreat. Well-sorted basin fines abruptly overlay the basal diamicts, reflecting a cessation of direct glacial input, although dispersed outsized clasts near the bottom have likely been deposited as ice-rafted debris. Frequently occurring plant fragments and fibres, and higher LOI values indicate that a milder climate prevailed

during the emplacement of the sediments. This study confirmed the assertion of Lang (2005) that a cryptotephra occurs in the sedimentary sequence of the Feldsee Bog. Future work will have to determine the major element composition of the tephra shards to unequivocally link it to the eruption of the Laacher See volcano. A mostly inorganic package at 5.00-4.75 m followed the first organic-rich package from 5.28 m to 5.00 m, possibly due to cooler climatic conditions. The emplacement of the gyttja-type sediments above 4.75 m depth occurred during persistently temperate conditions.

## 5.2 Radiocarbon ages

A total of nine macrofossils in sediment cores from the FSM coring site was radiocarbon dated. Table 1 summarises conventional $^{14}$C ages and calibrated radiocarbon ages. See also Fig. 5. Following the recommendation of Millard (2014), we hereinafter refer to 95% ranges of calibration.

Calibrating $^{14}$C ages of unidentified plant fragments sampled at decompacted depths of 5.61 m, 5.56 m, and 5.54 m resulted in overlapping ages of 15600—15210 cal. a BP, 15480—15040 cal. a BP, 15680—15240 cal. a BP, respectively. Sphagnum moss at a decompacted depth of 5.53 m yielded a modern $^{14}$C age. Leaf fragments at a depth of 5.43 gave an age of 13730—13170 cal. a BP. Sphagnum moss at a decompacted depth of 5.26 m yielded a $^{14}$C age of 720±30 a BP, whereas unidentified plant fibre at a decompacted depth of 5.24 m gave a modern $^{14}$C age. In contrast to the FSM-563, FSM-560, and FSM-558 samples, the appearance of sphagnum moss (FSM-553 and FSM-538 samples) and the plant fibre (FSM-536) was comparably fresh and thus the macrofossils are probably modern samples that were squeezed into older sediments during vibracoring. As these macrofossils apparently were modern samples, their $^{14}$C ages were not included in the age-depth model. Macrofossils sampled at a depth of 4.28 m gave almost identical ages of 9750—9530 cal. a BP and 9750—9490 cal. a BP, respectively (Table 1).

## 5.3 Luminescence ages

The IRSL ages of seven samples from the sediment cores from the FSM coring site ranged from 10.2±0.7 ka to 15.8±1.7 ka (Fig. 5). Table 4 summarises the luminescence data for the FSM coring site.

Whereas in the lower part of the sequence (samples FSM3, FSM4, and FSM5), the pIR ages are some 25% higher than the fading corrected IRSL and IR-50 ages, there is a good agreement between the different approaches for the two samples taken just above the cryptotephra (samples FSM1 and FSM2). The mean age for these latter samples is 11.4±0.4 ka, which is some 10-15 % lower than expected for sediments above the cryptotephra, provisionally classified as Laacher See Tephra (13006±9 cal. a BP; Reinig et al., 2021). FSM3 is also slightly to moderately underestimating the reported tephra age with estimates of 10.2±0.7 ka (IRSL) and 11.7±1.0 ka (IR-50). The ages for sample FSM4 (FG: IRSL = 12.7±1.0 ka, IR-50 = 13.2±1.0 ka; CG: 12.7±0.9 ka) overlap with the radiocarbon age of ca. 13.4 ka. The same applies to the basal sample FSM5, with ages of 14.0±1.2 ka (FG: IRSL), 14.2±1.2 ka (FG: IR-50), and 15.8±1.7 (CG: IRSL). Overall, there is a tendency of slightly lower

IRSL/IR-50 ages compared to independent age control through radiocarbon dating (Fig. 5). This is possibly explained by a slightly too low assumed sediment moisture due to compaction of the deposits with time, uncertainties related to the applied layer model, and/or correction of radioactive disequilibrium (timing of Uranium uptake). On the other hand, some the pIR ages appear slightly overestimated with might be related to small residual signal levels at the time of deposition. However, the consistent results determined for sample FSM4 indicate that fading correction is appropriate. Note that Fig. 5 only displays the IRSL ages for simplification.

## 5.4 Age-depth model

Figure 5 shows the age-depth model for the depth between 5.68 m and 4.00 m. Including all $^{14}$C and IRSL ages, and the reported age of the cryptotephra gave an agreement index of 0.1%, being well below the critical threshold of 60% advocated by Bronk Ramsey (2008). The IRSL ages for the FSM2, FSM3, and FSM3 samples were excluded during the second and subsequent runs, as individual agreement indices turned out to be lower than 60%. The final age-depth model gave an agreement index of 76.3%. The modelled age (mean age and 1σ uncertainty) and the 95% modelled age range for a decompacted depth of 5.68 m turned out to be 15640±420 cal. a BP (15.70±0.42 ka before 2010 CE) and 16470—15230 cal. a BP, respectively. Hence, the deposition of lake sediments commenced at 16470—15230 cal. a BP, thus providing the minimum age of ice-free conditions which will just post-date glacier recession from positions FS-02 and FS-03.

## 5.5 Beryllium-10 concentrations in rock samples from moraine boulders

The blank-corrected Beryllium-10 concentrations in moraine-boulder surfaces ranged from 92300±4200 to 161000±13000 atoms g$^{-1}$ quartz. Table 4 summarises the results of AMS measurements and blank-corrected $^{10}$Be concentrations. Figure 6 displays normalised $^{10}$Be concentrations with respect to the normalised error-weighted mean $^{10}$Be concentration. During AMS measurements on the FS-03b sample, $^{9}$Be currents were very low, decreasing from 0.1 to 1 µA during AMS measurements. Hence, the $^{10}$Be concentration in the sample from the boulder should be considered with greatest caution.

## 5.6 Regional $^{10}$Be production rate

Calibrating the spallogenic BFPR with the $^{10}$Be concentrations in the FS-02a, FS-02b, FS-02c, FS-03a, FS-03c, FS-03d, FS-03e and FS-03f moraine-boulder surfaces resulted in SLHL production rates between 3.61±0.11 and 3.65±0.11 $^{10}$Be g$^{-1}$ quartz a$^{-1}$ for the different scaling schemes and geomagnetic databases in CREp (Table 5).

The $^{10}$Be concentration in the FS-03b moraine-boulder surface was not used for production rate calibration due to low $^{9}$Be currents during AMS measurements. Correcting the $^{10}$Be concentrations for topographic shielding and the sample thickness led, in most cases, to minor shifts in the $^{10}$Be concentrations in samples. During the evaluation of the set of $^{10}$Be concentrations with the criterion of Peirce, the $^{10}$Be concentration in the sample from the FS-02d moraine boulder turned out to be an outlier. Hence, weighted mean concentrations were computed with the $^{10}$Be concentrations in the remaining

samples. In all cases, MSWD exceeded the critical value of one (see Martin et al., 2017 for further discussion). Following Martin et al. (2017), the standard error of the weighted mean and $\sqrt{MSWD}$ were added in quadrature to obtain the uncertainties of the weighted mean [10]Be concentrations. See Table 4 for the [10]Be concentration in moraine-boulder surfaces

that have been corrected for topographic shielding and the sample thickness and have been adjusted to SLHL.

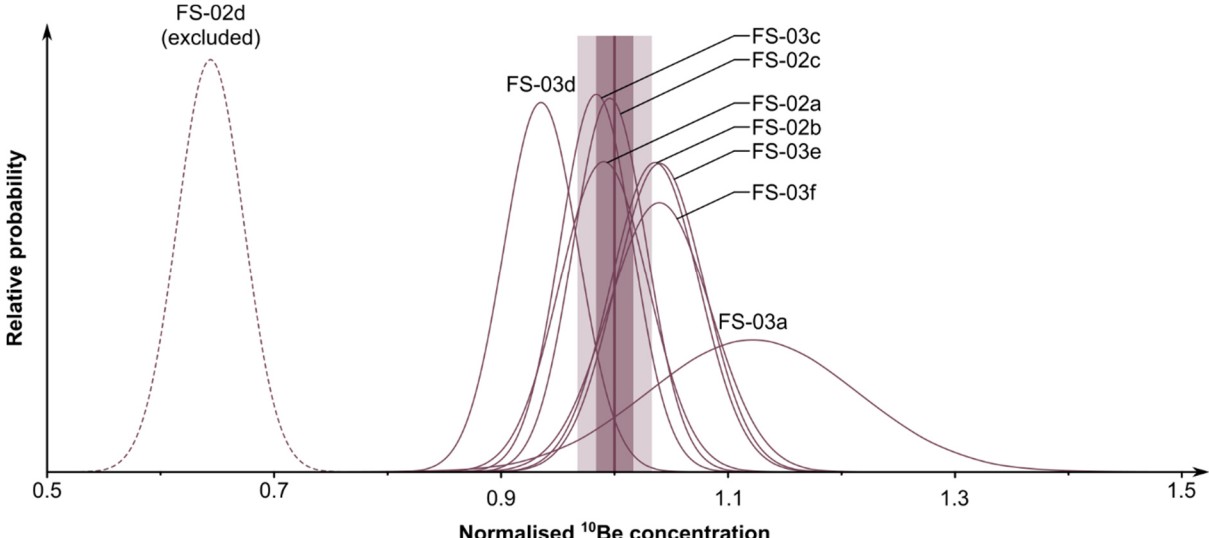

**Figure 6: Normalised [10]Be concentrations in rock samples from the study site. The [10]Be concentrations have been corrected for thickness and topographic shielding, scaled to SLHL using time-dependent 'Lm' scaling, and normalised to the error-weighted**

**mean [10]Be concentration utilised for production rate calibration. The purple curves are Gaussian approximations of [10]Be concentrations in individual samples. The vertical purple band and the vertical light purple band represent 1σ and 2σ uncertainties of the error-weighted mean [10]Be concentration, respectively. The vertical purple line corresponds to the error-weighted mean [10]Be concentration.**

The "baseline" production ranged from 3.61±0.11 to 3.65±0.11 atoms [10]Be g[-1] quartz a[-1] at SLHL for the different scaling schemes and geomagnetic databases in CREp (Table 5). These production rate resembled the production rate for 'Lm' (Nishiizumi et al., 1989; Lal, 1991; Stone, 2000; Balco et al., 2008) and 'St' (Stone, 2000) scaling (Table 6), calculated with version 3 of the online exposure age calculator, formerly known as the CRONUS-Earth online exposure age calculator (Balco et al., 2008).


Removing the snow cover bias at the study site led to 2.7% higher production rates between 3.70±0.12 and 3.75±0.12 atoms [10]Be g[-1] quartz a[-1] at SLHL. The snow-shielding- and postdepositional-weathering-corrected production rates ranged from 3.82±0.12 to 3.87±0.12 atoms [10]Be g[-1] quartz a[-1] at SLHL and were thus 6% higher than the "baseline" production rates. With 3.90±0.12 to 3.95±0.12 atoms [10]Be g[-1] quartz a[-1] at SLHL, the postdepositional-weathering-, snow-cover-, and forest-

cover-corrected production rates were 8% higher than the "baseline" production rates (Table 5).

**Table 5: The spallogenic $^{10}$Be SLHL BFPR (atoms $^{10}$Be g$^{-1}$ quartz a$^{-1}$) for different scaling schemes and geomagnetic databases available in CREp (Martin et al., 2017).**

| Scaling scheme | Time-dependent Lal/Stone scaling (Nishiizumi et al., 1989; Lal, 1991; Stone, 2000; Balco et al., 2008) | | | LSD (Lifton et al., 2014) | | |
|---|---|---|---|---|---|---|
| Geomagnetic database | Atmospheric $^{10}$Be-based VDP (Muscheler et al., 2005) | LSD framework (Lifton et al., 2014) | Lifton VDM 2016 (Lifton, 2016) | Atmospheric $^{10}$Be-based VDP (Muscheler et al., 2005) | LSD framework (Lifton et al., 2016) | Lifton VDM 2016 (Lifton, 2016) |
| "Baseline" production rate | 3.64±0.11 | 3.61±0.11 | 3.62±0.11 | 3.65±0.11 | 3.63±0.11 | 3.63±0.11 |
| Production rate (snow shielding bias removed) | 3.74±0.12 | 3.70±0.12 | 3.72±0.12 | 3.75±0.12 | 3.73±0.12 | 3.73±0.12 |
| Production rate (snow shielding and postdepositional weathering bias removed) | 3.77±0.12 | 3.73±0.12 | 3.75±0.12 | 3.78±0.12 | 3.76±0.12 | 3.76±0.12 |
| Production rate (snow shielding, postdepositional weathering, and forest cover bias removed) | 3.85±0.12 | 3.81±0.12 | 3.82±0.12 | 3.86±0.12 | 3.83±0.12 | 3.83±0.12 |

**Table 6: The spallogenic $^{10}$Be SLHL BFPR for two scaling schemes in version 3 of the online exposure age calculator, formerly known as the CRONUS-Earth online exposure age calculator (Balco et al., 2008).**

| Scaling | SLHL production rate (atoms g$^{-1}$ quartz a$^{-1}$) |
|---|---|
| 'Lm' (Nishiizumi et al., 1989; Lal, 1991; Stone, 2000; Balco et al., 2008) | 3.65±0.20 |
| 'St' (Stone, 2000) | 3.62±0.20 |

**5.7 Beryllium-10 CRE ages**

Calculating the $^{10}$Be CRE ages of moraine-boulder surfaces with the BFPR resulted in ages between 10.2±0.5 and 17.6±1.5 ka. CRE ages calculated with the Chironico landslide spallogenic SLHL production rate lay between 9.1±0.5 and 15.7±1.3 ka. See Table 6 and Fig. 7 for CRE ages of moraine boulders, landform ages, and reduced $\chi^2$ values.


**Table 7: CRE ages of sampling surfaces on moraine boulders computed with the Black Forest (this study) and Chironico landslide production rates (Claude et al., 2014). *Classified as an outlier. **Due to low $^9$Be counts during AMS measurements, the $^{10}$Be concentration in the sample is not reliable and, therefore, no CRE age was computed.**

| Ice-marginal position | Boulder | Beryllium-10 CRE age (ka before 2010) and internal uncertainty in ka [external uncertainty (ka) in parentheses] | | Age difference (%) | Landform age (ka before 2010) and internal uncertainty in ka [external uncertainty (ka) in parentheses] | | Reduced $\chi^2$ | | | |
|---|---|---|---|---|---|---|---|---|---|---|
| | | | | | | | Before the exclusion of outliers | | After the exclusion of outliers | |
| | | BFPR | Chironico landslide production rate | | BFPR | Chironico landslide production rate | BFPR | Chironico landslide production rate | BFPR | Chironico landslide production rate |
| FS-01 | FS-01a | 14.94±0.50 (0.66) | 13.30±0.46 (0.56) | 12.3 | 14.94±0.50 (0.66) | 13.30±0.46 (0.56) | - | - | - | - |
| FS-02 | FS-02a | 15.52±0.59 (0.74) | 13.84±0.53 (0.62) | 12.1 | 15.58±0.03 (0.47) | 13.88±0.02 (0.34) | 32.5 | 32.5 | 0.0 | 0.0 |
| | FS-02b | 15.58±0.57 (0.72) | 13.88±0.51 (0.60) | 12.2 | | | | | | |
| | FS-02c | 15.62±0.48 (0.67) | 13.91±0.44 (0.54) | 12.3 | | | | | | |
| | FS-02d | 10.24±0.45 * (0.54) | 9.12±0.40 * (0.46) | 12.3 | | | | | | |
| FS-03 | FS-03a | 17.59±1.39 (1.48) | 15.65±1.23 (1.29) | 12.3 | 16.06±0.42 (0.64) | 14.32±0.37 (0.51) | 2.6 | 2.5 | 1.4 | 1.3 |
| | FS-03b | - ** | - ** | - | | | | | | |
| | FS-03c | 15.55±0.48 (0.66) | 13.86±0.44 (0.54) | 12.2 | | | | | | |
| | FS-03d | 14.75±0.50 (0.66) | 13.13±0.45 (0.55) | 12.3 | | | | | | |
| | FS-03e | 16.34±0.59 (0.76) | 14.56±0.53 (0.63) | 12.2 | | | | | | |
| | FS-03f | 16.34±0.68 (0.83) | 14.55±0.60 (0.69) | 12.3 | | | | | | |

Table 6 reveals that the ages calculated with the BFPR turned out to be, on average, 11.0% higher than those computed with
       the Chironico landslide production rate. We hereinafter only comment on the ages and 1σ uncertainties derived with the
       BFPR. External uncertainties are given in parentheses.

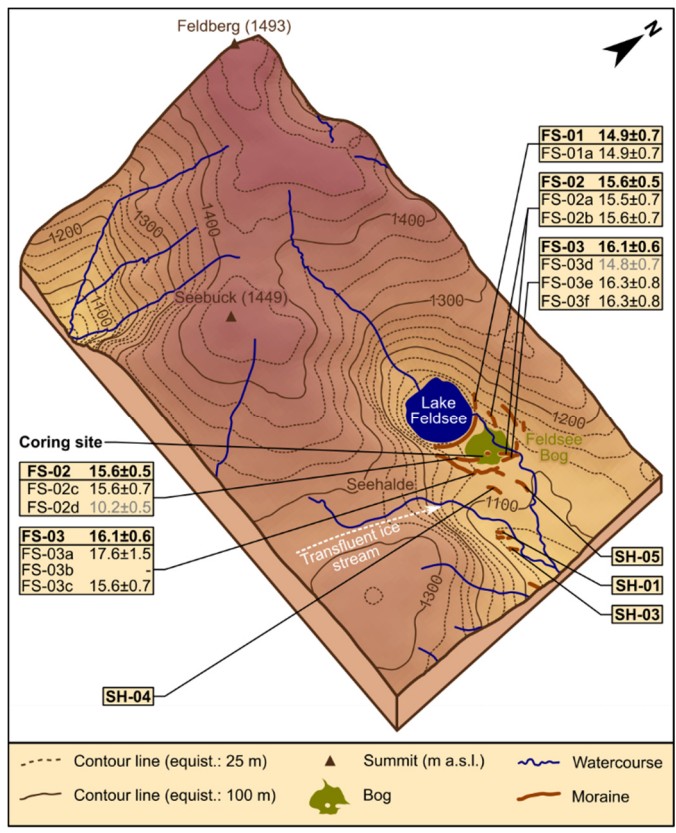

**Figure 7: Map of moraines at the study site and the immediate surroundings. CRE ages and associated external uncertainties of
       moraine-boulder surfaces are given in ka before 2010 CE and ka, respectively. Bold ages are landform ages and landform age
       uncertainties (i.e., external uncertainties). Ages in grey were classified as outliers. See NASA Jet Propulsion Laboratory (2013) for
       a description of the DEM in the background.**

The FS-03a, FS-03c, FS-03d, FS-03e, and FS-03f boulders on the ice-marginal moraine at position FS-03 yielded ages of
       17.59±1.39 (1.48) ka, 15.55±0.48 (0.66) ka, 14.75±0.50 (0.66) ka, 16.34±0.59 (0.76) ka, and 16.34±0.68 (0.83) ka,
       respectively. As mentioned above, the [10]Be concentration in the sample from the FS-03b boulder should not be regarded
       reliable due to low [9]Be currents during Be AMS measurements and, thus, no age was calculated. The age of the FS-03d
       boulder (14.75±0.50 (0.66) ka) was classified as an outlier. The remaining ages gave a landform age of 16.06±0.42 (0.64) ka
(Fig. 7 & Table 7).

The FS-02a, FS-02b, FS-02c and FS-02d boulders on the moraine at position FS-02 gave ages of 15.52±0.59 (0.74) ka, 15.58±0.57 (0.72) ka, 15.62±0.48 (0.67) ka, and 10.24±0.45 (0.54) ka, respectively. After the exclusion of the outlying age of the FS-02d boulder (10.2±0.5 (0.5) ka), the remaining ages yielded a landform age of 15.58±0.03 (0.47) ka (Table 6 & Fig. 7).

The only sufficiently large and stable boulder on the ice-marginal moraine at position FS-01 was exposure dated to 14.94±0.50 (0.66) ka (Table 6 & Fig. 7).

Overall, the presented ages were, apart from a few outliers, internally consistent, and the landform ages complied with the stratigraphy. With 15.4±0.8 ka, 15.4±0.8 ka, and 15.4±0.7 ka, the ages of the FS-02a, FS-02b, and FS-02c boulders on the moraine at position FS-02 were remarkably consistent (Reduced $\chi^2$: 0.0; Table 6).

## 5 Discussion

The determination of concentration of in situ accumulated cosmogenic [10]Be in boulders on moraines in the Feldsee Cirque and applying radiocarbon dating, tephrochronology, and IRSL dating to stratigraphically younger lake sediments allowed us to establish the internally robust BFPR production rate, the first [10]Be production rate for the mid-elevation mountain ranges of central Europe. In Sect. 5.1, we discuss the internal robustness of this production rate. We then evaluate previously suggested corrections of CRE ages for postdepositional weathering and snow shielding (Sect. 5.2). In Sect. 5.3, we finally discuss the newly calibrated BFPR in the European context. A detailed discussion of the implications of this study for the Late Pleistocene glacial history of mid-elevation mountain ranges would go beyond the scope of this paper and will be discussed in future work.

### 5.1 Internal robustness of the BFPR

The modelled basal age of the lake sediments above the moraine at the ice-marginal position FS-02 (mean age and 1σ uncertainty: 15640±420 cal. a BP) underpins the SLHL BFPR. Radiocarbon dating of macrofossils near the base of the lake sediments, i.e., at decompacted depths of 5.61, 5.56, and 5.54 m resulted in consistent ages of 15600—15210 cal. a BP, 15480—15040 cal. a BP, and 15680—15240 cal. a BP, respectively. These ages agree with the concordant IRSL ages of the FSM5-CG and FSM5-FG samples, luminescence dated to 15.8±1.7 ka and 14.0±1.2 ka, respectively. Due to consistent ages obtained with two independent dating methods, we regard the modelled basal age of the lake sediments (15640±420 cal. a BP) as reliable.

The calibration of the BFPR was based on two implicit assumptions. First, we deemed the modelled basal age at the FSM coring site a minimum age for the ice-marginal moraines at positions FS-03 and FS-02. The time lag between glacier

recession from the moraines at these positions and the deposition of the macrofossils in the former proglacial lake remains unknown. Unfortunately, no bog exists downstream from the position FS-03 that would have allowed us to determine a maximum radiocarbon or luminescence age of the moraines at position FS-03. We were thus unable to derive the lowest possible $^{10}$Be production rate at the study site.

Second, we implicitly assumed that all sampled moraine boulder surfaces had already been exposed to cosmic radiation when the macrofossils were deposited at the coring site. Previous work (Heyman et al., 2016) has revealed that larger boulders on moraines are more likely to have been continuously exposed to cosmic radiation than smaller boulders. Plotting the boulder height versus the normalised $^{10}$Be concentrations, i.e., the scaled $^{10}$Be concentrations, revealed the absence of a relationship between these variables (Fig. 8a). The low and statistically insignificant coefficient of determination ($R^2 = 0.03$; $p = 0.84$) contradicts the hypothesis that post-depositional and post-stabilisation exhumation is a serious issue at the study site.

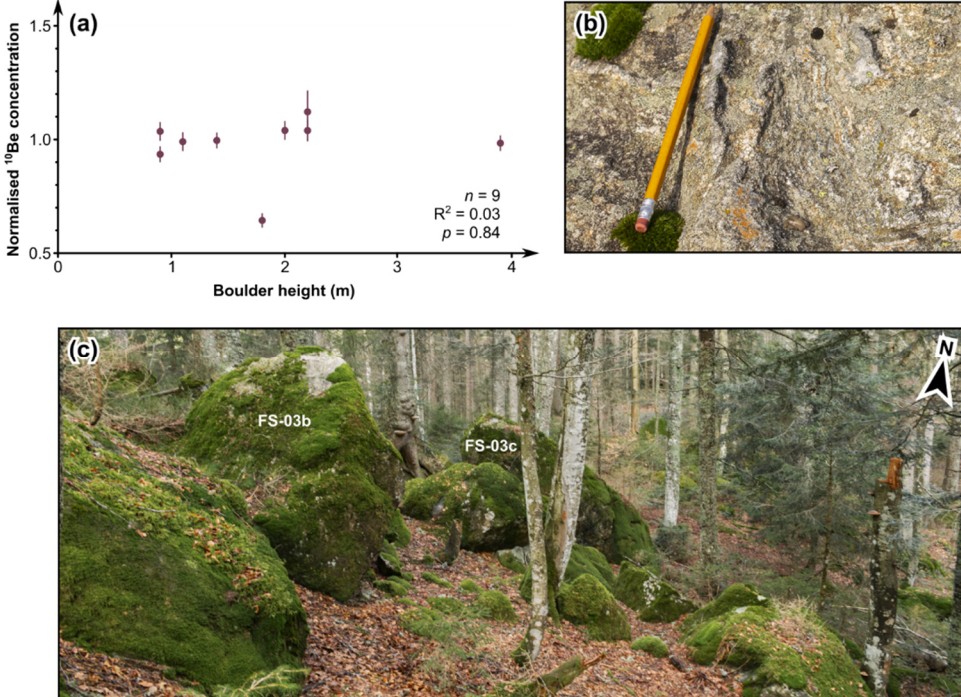

**Figure 8: (a) Boulder heights versus normalised $^{10}$Be concentrations. (b) Protruding quartz veins on a random boulder on the ice-marginal moraine at position FS-03. (c) Boulder-rich ice-distal side of the moraine at position FS-03 where the large FS-03b and FS-03c boulders were sampled. All photos: FMH.**

Previous work has also demonstrated that landform stability critically determines CRE ages of moraine-boulder surfaces and that the position of the boulders on the landform, i.e., the moraine crest, the ice-proximal slope, or the ice-distal slope, does

not matter (cf., Tomkins et al., 2021). Figure 8c shows a photo of the ice-distal side of the moraine at position FS-03 where the FS-03a, FS-03b, and FS-03c boulders have been sampled. Note that this part of the moraine almost entirely consists of boulders. We therefore deem significant post-depositional reworking of the moraine unlikely. It should be noted that, despite their high age, the moraines at positions FS-03 and FS-02 still have distinct crests. We interpret the comparably high landform stability as a further indication that the sampled moraine boulders had already been exposed to cosmic radiation when the deposition of lake sediments at the FSM drilling site began.

## 5.2 Evaluation of previously suggested corrections for snow shielding and postdepositional weathering

Due to the lack of a regional calibration site for the mid-elevation mountains of central Europe, CRE ages presented in previous studies were determined with SLHL $^{10}$Be production rates at localities outside this region and subsequently corrected for snow shielding and weathering (e.g., Mercier et al., 1999; Reuther, 2007; Hofmann et al., 2022, 2024b). Note that this approach is technically wrong, as SLHL production rates at calibration sites have their own bias with regard to weathering, snow shielding, and other factors. Previous studies thus corrected, to some extent, twice for snow shielding and weathering.

Irrespective of this discussion, the "baseline" BFPR offers the unique opportunity to critically assess whether these corrections were sufficient. For example, Hofmann et al. (2022, 2024b) calculated CRE ages for the southern Black Forest with the SLHL $^{10}$Be Chironico landslide production rate (Claude et al., 2014), as this calibration site was the closest calibration site to the southern Black Forest. These authors corrected CRE ages for the supposedly stronger snow shielding bias in the Black Forest with an additional snow shielding factor (~0.974), deduced from meteorological data obtained between 1961 CE and 1990 CE at the weather station near Lake Titisee, located at about 850 m a.s.l. and about 10 km to the NE of the study area. The correction for the supposedly stronger postdepositional weathering bias was based on a weathering rate for homogenous paragneiss from the Bavarian Forest (0.24 cm ka$^{-1}$; recalculated from Reuther, 2007). Correcting the ages resulted in an age shift of about 5% (Hofmann et al., 2022, 2024b).

Figure 9 shows the uncorrected and snow-shielding- and weathering-corrected $^{10}$Be ages for the study site (Chironico landslide SLHL production rate) as well as those computed with the BFPR. Figure 9 reveals that the uncorrected CRE ages do, apart from the ages for the FS-03a boulder surface, not overlap with those calculated with the BFPR within external uncertainties. Three possible explanations for the age differences appear possible: (i) previous studies have corrected the ages sufficiently for snow cover and postdepositional weathering but have ignored other factors that do affect CRE ages, such as soil cover on the boulders at the study site. (ii) Previous studies might have simply corrected the ages insufficiently for snow cover and postdepositional weathering, and other factors do not play an important role. (iii) The correction for snow cover and postdepositional weathering was insufficient and previous studies have ignored other important factors. As the CRE ages computed with the BFPR align with independent age control, we propose that the BFPR should be used in

future dating studies on the mid-elevation mountains of Central Europe and that previous age estimates should be reconciled with the BFPR.

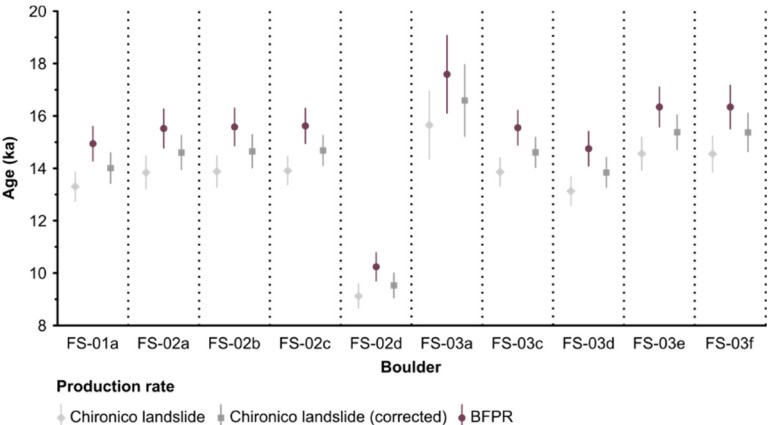

**Figure 9: CRE ages of moraine-boulder surfaces and external CRE age uncertainties, calculated with the Chironico landslide**
**SLHL ¹⁰Be production rate (Claude et al., 2014) and the SLHL ¹⁰Be BFPR. Note that all ages were computed in CREp (Martin et al., 2017) with the following parameters: time-dependent Lal/Stone scaling (Nishiizumi et al., 1989; Lal, 1991; Stone, 2000; Balco et al., 2008), the ERA atmosphere model (Uppala et al., 2005), and the atmospheric ¹⁰Be-based geomagnetic database of Muscheler et al. (2005).**

### 5.3 Comparison with previously published European production rates

We hereinafter discuss the SLHL ¹⁰Be BFPR in the context of previously published European production rates (Table 7). Note that all SLHL production rates in CREp mentioned in the following paragraphs refer to time-dependent Lal/Stone scaling (Nishiizumi et al., 1989; Lal, 1991; Stone, 2000; Balco et al., 2008), the ERA atmosphere model (Uppala et al., 2005), and the atmospheric ¹⁰Be-based geomagnetic database (Muscheler et al., 2005).

The internally robust "baseline" SLHL ¹⁰Be BFPR (3.64±0.11 atoms ¹⁰Be g⁻¹ quartz a⁻¹ at SLHL) does, apart from the SLHL ¹⁰Be production rates at the Russenes and Grøtlandsura rock avalanches (Fenton et al., 2011), not overlap at the 1σ level with other European SLHL ¹⁰Be production rate in CREp (Fig. 8a). Most notably, the SLHL ¹⁰Be BFPR is 11.2% and 11.4% lower than the ¹⁰Be Chironico landslide production rate (4.10±0.10 atoms ¹⁰Be g⁻¹ quartz a⁻¹ at SLHL) and the canonical global production rate (4.11±0.19 atoms ¹⁰Be g⁻¹ quartz a⁻¹ at SLHL) in CREp, respectively. However, the BFPR overlaps
with the global production rate and several regional production rates at the 2σ level (Fig. 8b). The agreement at the 2σ level with the global SLHL ¹⁰Be production rate in CREp is due to the relatively large 1σ error of this production rate (0.19 atoms ¹⁰Be g⁻¹ quartz a⁻¹ at SLHL).

**Table 8: The BFPR and previously published European [10]Be production rates in CREp (Martin et al., 2017). Note that all SLHL production rates refer to time-dependent Lal/Stone scaling (Nishiizumi et al., 1989; Lal, 1991; Stone, 2000; Balco et al., 2008), the ERA atmosphere model (Uppala et al., 2005), and the atmospheric [10]Be-based geomagnetic database (Muscheler et al., 2005). We did not include the Rannoch Moor production rate, as this production rate is not commonly accepted (cf., Lowe et al., 2019, and references therein). See Fig. 8b for the location of the sites.**

| Calibration site | Coordinates (WGS 1984 coordinate reference system) | | Elevation (m a.s.l.) | Age (ka before 2010 CE) | Beryllium-10 production rate (atoms [10]Be g$^{-1}$ quartz a$^{-1}$ at SLHL) | Reference |
|---|---|---|---|---|---|---|
| | Latitude | Longitude | | | | |
| Feldsee Cirque (Germany) | 47.8706 °N | 8.0375 °E | 1107 | 15.70±0.42 | 3.64±0.11 | This study |
| Chironico landslide (Switzerland) | 46.4155 °N | 8.8524 °E | 761 | 13.38±0.11 | 4.10±0.10 | Claude et al. (2014) |
| Mount Billingen 6 (Sweden) | 58.5189 °N | 13.7294 °E | 119 | 11.53±0.11 | 4.32±0.18 | Stroeven et al. (2015) |
| Mount Billingen 1 (Sweden) | 58.5341 °N | 13.7597 °E | 95 | 10.73±0.35 | 4.14±0.30 | |
| Mount Billingen 3 (Sweden) | 58.5003 °N | 13.6383 °E | 106 | 11.11±0.20 | 4.25±0.30 | |
| Halsnøy moraine (Norway) | 59.7808 °N | 5.79079 °E | 77 | 11.64±0.10 | 4.43±0.08 | Goehring et al. (2012) |
| Oldedalen rock avalanche (Norway) | 61.6667 °N | 6.8150 °E | 127 | 6.06±0.11 | 4.23±0.14 | |
| Grøtlandsura rock avalanche (Norway) | 68.9112 °N | 17.5224 °E | 71 | 11.47±0.11 | 3.48±0.10 | Fenton et al. (2011) |
| Russenes rock avalanche (Norway) | 69.2135 °N | 19.4731 °E | 109 | 10.99±0.08 | 3.84±0.13 | |
| Kyleakin Pass (Scotland) | 57.2206 °N | 5.7243 °W | 309 | 11.75±0.30 | 4.46±0.26 | Borchers et al. (2016) |
| Coire Mhic Fearchair (Scotland) | 57.2408 °N | 5.9727 °W | 442 | | 4.23±0.12 | |
| Coire nan Arr (Scotland) | 57.4141 °N | 5.6438 °W | 139 | | 4.27±0.11 | |
| Maol Chean Dearg (Scotland) | 57.4874 °N | 5.4493 °W | 521 | | 4.20±0.11 | |

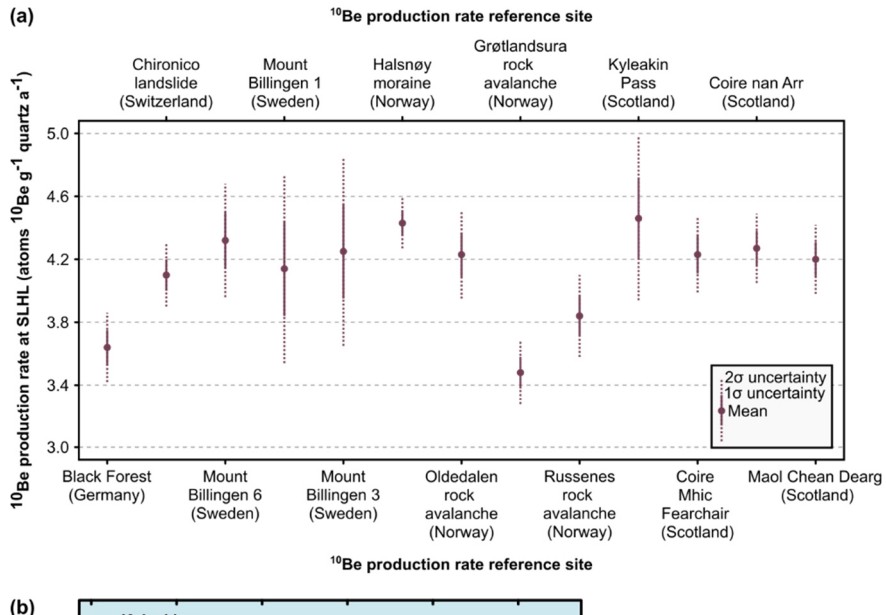

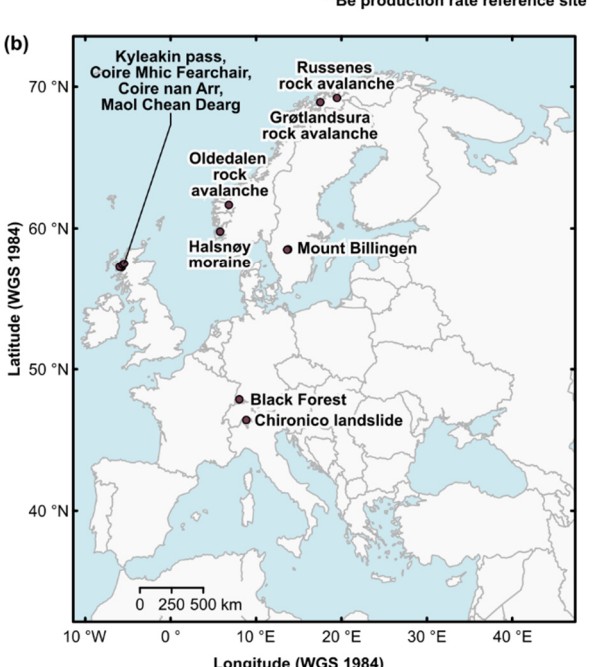

**Figure 10: (a) European ¹⁰Be production rates available in CREp (Martin et al., 2017). We did not include the Rannoch Moor ¹⁰Be production rate in this compilation, as this production rate is not commonly accepted (cf., Lowe et al., 2019, and references therein). (b) Map of the calibration sites mentioned in panel (a).**

## 5.4 Explanations for the anomalously low BFPR

Overlapping SLHL production rates (at the 1σ level) for the Chironico landslide and the Black Forest would have additionally required that (i) changes in the geomagnetic field simultaneously affected both sites (ii) postdepositional weathering rates were similar, (iii) the snow cover on the sampling surfaces during winter was comparable, (iv) shielding by

trees reduced the cosmic ray flux at both sites in a similar magnitude, and that (v) other factors, such as soil cover on sampling surfaces, played a similar role during the exposure of the boulders.

Due to the comparably short distance between the Chironico landslide and the Feldsee Bog (~170 km), changes in the geomagnetic field probably do not account for the offset. The postdepositional weathering bias at the study site was probably stronger due to the presence of soils, mosses, and shrubs on most of the sampled boulders that slowed down the production rate (cf., Ahnert, 2009) and resulted in higher weathering rates (cf., Dunai, 2010). It should be noted that the sampled boulder surfaces on the Chironico landslide "did not show any obvious signs of damage or spalling" (Claude et al., 2014:

277). As we observed protruding quartz veins on three moraine boulders at the study site, we argue that the higher rate of postdepositional weathering at the study site likely partly explains the difference in production rates. The snow cover bias at the study site was probably stronger than at the Chironico landslide, as the Feldsee Bog is situated in a sheltered position, at a much higher elevation, and at a higher latitude. To the best knowledge of the authors, no data on the duration and thickness of the snow cover during winter are available from a weather station close to the Chironico landslide. We were thus unable

to confirm this assumption with quantitative data. The sampled boulders at both the Feldsee Bog and the Chironico landslide were situated in forested areas and, thus, the forest canopy probably slowed down the effective production rates at both sites in a similar order of magnitude. As the landslide is located at both a lower latitude and elevation, the forest cover bias at the latter site should have been stronger. In contrast to the boulders at the study site, the boulders on the Chironico landslide sampled by Claude et al. (2014) were devoid of soil, mosses, and shrubs. We thus conclude that a stronger snow-cover- and

postdepositional-weathering bias in the study area, and the presence of soil, mosses, and shrubs on the moraine-boulder surfaces at the study site explain the relatively low BFPR.

**6 Conclusions**

Applying a multi-method approach to ice-marginal moraines and stratigraphically younger lake sediments in the southern Black Forest yielded consistent geochronological data and allowed for deriving the SLHL $^{10}$Be BFPR, the first $^{10}$Be

production rate for the mid-elevation mountain ranges of central Europe. The BFPR turned out to be lower than most European production rates and about 11% lower than the canonical global $^{10}$Be production rate in CREp. We attribute this offset to a strong influence of snow shielding, postdepositional weathering, and soil, moss, and shrub cover at the study site. Assessing previously suggested corrections for snow shielding and postdepositional weathering revealed that these corrections were probably too weak. Due to a lower 1σ error (0.11-0.12 atoms $^{10}$Be g$^{-1}$ quartz a$^{-1}$ at SLHL) than the 1σ

uncertainty of the canonical global production rate in CREp (0.19 atoms $^{10}$Be g$^{-1}$ quartz a$^{-1}$ at SLHL), the application of BFPR will likely allow for calculating more accurate CRE ages for settings that resemble the study site. Among other factors, these localities must be similar in terms of weathering rates, snow cover biases and tree shielding.

This study shows that ages generated with three independent geochronological methods, namely radiocarbon dating, IRSL dating, and tephrochronology, can successfully be interrogated in a Bayesian approach for the establishing an age-depth model for lake sediments. If such a model relies on multiple, independent geochronological methods, ages with higher uncertainties, such as IRSL ages, can also be included. The integration of these ages then leads to improvements to the age-depth model and increases the precision of the production rate. However, integrating a comparably imprecise geochronological method only works if this method is applied in tandem with a precise dating method, such as radiocarbon dating. If this is not the case, the low precision of the unmodelled ages results in a large uncertainty in the age model, which is reflected in a large error in the production rate.

**Data availability**

Watercourses and lakes in Fig. 6 are available at LUBW (2022a) and LUBW (2022b), respectively. The DEM in Figs. 1 and 6 (NASA Jet Propulsion Laboratory, 2013) can be downloaded in the Earth Explorer of the United States Geological Survey (USGS; https://earthexplorer.usgs.gov, last access: 5 December 2023). See DWD (2023) for meteorological data for the Feldberg weather station. We included all other relevant data supporting this study in the supplement and submitted the calibration data to the ICE-D, the informal cosmogenic-nuclide exposure-age database.

**Team list**

ASTER Team: Georges Aumaître, Karim Keddadouche, and Fawzi Zaidi

**Author contribution**

FP and FMH designed the study. FMH and AF undertook the fieldwork. FMH, CR, LG, MaS, MeS, AF, and LL conducted laboratory work. The ASTER Team performed the AMS measurements. FMH, CR, LG, MaS, MeS, AF, and FP processed and interpreted the data. FMH drafted the manuscript and the figures with contributions from FP, LG, and CR. All authors contributed to the final version of the manuscript.

**Competing interests**

The authors have declared that there are no competing interests.

**Acknowledgements**

We thank Florian Rauscher for his help during fieldwork and for his support during sample preparation for CRE dating. William McCreary III supported us during the coring campaign at the Feldsee Bog. The authors thank Harri Geiger for his help with the identification of the cryptotephra. The forestry department of the Breisgau-Hochschwarzwald district kindly provided a forest access permit. We thank the nature protection department of *Regierungspräsidium Freiburg* and the Feldberg Natural Reserve, particularly Clemens Glunk and Achim Laber, for the permission to sample moraine-boulder surfaces for CRE dating. Matthias Geyer (Geotourist Freiburg) kindly provided the oblique aerial photo of the study site taken from a gilder. We cordially thank the State Geological Survey of Baden-Württemberg (LRGB), particularly Ulrike Wieland-Schuster, for providing LiDAR data of the Baden-Württemberg State Agency for Spatial Information and Rural Development (LGL) in a collaborative way. We thank the reviewers and the associate editor, Greg Balco, whose critical and constructive comments resulted in significant improvements to this paper.

**Financial support**

This research was undertaken while Felix Martin Hofmann was in receipt of a PhD studentship of *Studienstiftung des Deutschen Volkes*. This study was financially supported by the German Research Foundation (DFG) through the 'Geometry, chronology and dynamics of the last Pleistocene glaciation of the Black Forest', 'Chronology of the glaciation of the southern Black Forest after the Late Pleistocene glaciation maximum', and 'Gerät für Lumineszenz-Messungen und Datierungen' projects granted to FP (project numbers: 426333515, 516126018, and 282256512, respectively). The French AMS national facility ASTER at CEREGE is supported by the INSU/CNRS, the ANR through the 'Projets thématiques d'excellence' programme for the 'Equipements d'excellence' ASTER-CEREGE action and IRD.

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
