# Peer review of "Regional Beryllium-10 production rate for the mid-elevation mountainous regions in central Europe, deduced from a multi-method study of moraines and lake sediments in the Black Forest"

_Geochronology, 2023_

## Author Comment (AC1)

**Reply to the comments of Anonymous Referee #1 to the manuscript entitled 'Local Beryllium-10 production rate for the mid-elevation mountainous regions in Central Europe, deduced from a multi-method study of moraines and lake sediments in the Black Forest'**

Dear reviewer,

We thank you for your thoughtful and critical comments that resulted in considerable improvements to the manuscript. We thoroughly considered all comments and revised the manuscript accordingly. For responses to the comments, see the table below. We hope that the manuscript will be accepted for publication in its revised form.

Thank you very much for your kind consideration.

On the behalf of all co-authors,

Felix Martin Hofmann

| Line, Figure, or Table | Reviewer comment | Authors' reply |
|---|---|---|
| - | **Potential errors in data reporting:** After a thorough review of the data presented in this manuscript, I believe I might have found some mathematical errors that I strongly encourage the authors to double check, mainly in the calculation of cosmogenic $^{10}$Be concentrations that I found to be 2-3% too low compared to my own calculations. See below for more information on that. Moreover, the authors remove from consideration three samples in the calibration dataset - one sample is an extreme value that could rightfully warrant removal in my opinion, one sample they argue had sample measurement issues and should be disregarded, but the last one (FS-01a) seemingly does not have any explanation from the authors. My only thought is that perhaps it was removed because the sampled boulder was situated on a moraine stratigraphically above the bog and the authors only wanted to consider the modeled radiocarbon date as a minimum age constraint. However, the normalized concentration that I calculated for this boulder is nearly identical to the other samples within the resolution of the dating method so I am not sure it should be removed. In fact, the radiocarbon constraint from the bog could conceivably be considered a maximum age constraint for the younger moraine. I would like to see either a much clearer explanation as to why they removed this sample from consideration, or I feel the authors should reconsider including it in the calibration dataset. | We carefully checked our calculations for potential errors. See the attached table for further details. We did not include the Beryllium-10 concentration in the sampling surface on the FS-01a boulder in the calibration dataset, as moraine formation at position FS-01 might have postdated the onset of the deposition of lake sediments at the Feldsee Bog. We added this information to the revised methods section:

"We also sampled the surface of the FS-01a gneiss boulder on the moraine at position FS-01a for age calculations. However, we did not include the sample in the calibration dataset, as moraine formation might have post-dated the onset of deposition of lake sediments at the FSM coring site (Fig. 3b)."

We agree that the basal age at the FSM coring site provides a maximum age for the moraine at position FS-01. Unfortunately, only one boulder was available for sampling on this landform. |

| Line, Figure, or Table | Reviewer comment | Authors' reply |
|---|---|---|
| - | **Sediment coring approach::** Although I am not requesting the authors specifically address this if it is outside their scope, I am very curious about their (and previous studies') sediment coring approach. As far as I can tell, the authors only measured radiocarbon on macrofossils collected in one sediment core even though there have been 13 cores recovered from this bog according to the text and figure 4. Is there a specific reason the authors noted all the other cores in this manuscript even though I assume they are not reporting radiocarbon dates from any of the other cores? Have the authors (or original core collectors) recovered macrofossils in any other cores to help corroborate the results from the one core presented here? I recommend the authors either shift focus away from, and perhaps even omit mention of the other cores, or present radiocarbon data from the cores if they exist to help support the reported dates. Afterall, the independent age constraint essentially hinges on just 3 three radiocarbon constraints from one section in one core in the bog. | Lang et al. (1984) obtained 13 sediment cores from the Feldsee Bog. To the best knowledge of the authors, the cores do not exist anymore. Therefore, the authors of this study undertook a coring campaign to retrieve new sediment cores. To make this clear, we reformulated the beginning of the methods section as follows:

"To the best knowledge of the authors, the cores obtained by Lang et al. (1984) do, unfortunately, not exist anymore. We thus obtained sediment cores at the FSM coring site during fieldwork in 2021 CE." |
| - | **Figures and tables:** In terms of general comments, I feel that readers would benefit from revisions to some of the figures and tables in the paper. See specific comments below. I also recommend the authors include one additional figure of normalized concentrations from every sampled boulder so readers can more clearly assess the measurement results from each boulder relative to each other (see my comments on table 6 and suggested additional figure), and one additional figure plotting normalized concentrations versus boulder heights for all samples. | We revised some of the figures and tables according to the suggestion of the reviewer. For example, the sampled boulders were added to Fig. 3. In addition, we added the two suggested figures to the manuscript in order to improve the clarity of the text. |

| Line, Figure, or Table | Reviewer comment | Authors' reply |
|---|---|---|
| - | **Alternative explanation for the relatively low reference production rate:** The authors identify (Line 250) an important point about post-depositional disruption and exhumation impacting $^{10}$Be accumulation. Because this PR calibration site is so much lower than other sites, it forces me to wonder if there really is an exhumation/stabilization issue going on here. I recommend the authors dedicate more discussion around the morphology of the moraines. In my experience, boulders embedded in moraines (as opposed to fully atop or even better, clast-supported) are likelier candidates for exhumation issues, especially if there is local, late Pleistocene seismicity. Additionally, the authors observe a lack of protruding quartz veins on their boulders (Line 610-611), which in my experience could mean the boulders were shielded from weathering and potentially exhumed long after deposition. Can exhumation issues be truly ruled out here? As written, I am not fully convinced, and I recommend the authors discuss this issue in more detail. | The first author of the manuscript undertook fieldwork and carefully double-checked the sampled boulders for protruding quartz veins. In contrast to the sampling campaign, he observed protruding quartz veins on three moraine boulders (on the FS-02b boulder and on two random moraine boulders at position FS-03). The protruding quartz vein on the FS-02b boulder had a height of 1 cm. See Fig. 8b for a photo of the exposed quartz veins on a random moraine boulder at position FS-03.

Regarding landform stability, it should be noted that the sampled moraine boulders at position FS-03 were large (see Fig. 10c). Some parts of the moraine at position FS-03a only consisted of large boulders (Fig. 8c), such as the portion of the landform where the FS-03a, FS-03b, and FS-03c boulders have been sampled. We comment on the moraines' morphology in one paragraph in the revised methods section:

"Since the study of Tomkins et al. (2021) demonstrated that landform stability mainly influences the scatter in age distributions from moraines (and thus in 10Be concentrations), only well-embedded boulders were selected to avoid underestimated $^{10}$Be concentrations due to boulder rotating as well as post-depositional and post-stabilisation exhumation. As the moraine at position FS-03 consisted of clast-supported diamicts and some parts of the moraine were solely composed of boulders, identifying stable and large boulders proved to be straightforward. The same was true for the moraine at position FS-02 although this landform consisted of matrix-supported diamicts. Identifying large and stable boulders on the moraine at position FS-01 turned out to be difficult, as this landform consisted of matrix-rich diamicts and since the moraine exhibited only a few boulders. We thus only sampled one large boulder." |

| Line, Figure, or Table | Reviewer comment | Authors' reply |
|---|---|---|
| | | "Hofmann and Konold (2023) mapped a kettle on the proximal side of the moraine at position FS-03 and on the moraine at position FS-02 in the centre of the Feldsee Bog (Fig. 3), pointing to paraglacial reworking and delayed moraine stabilisation (cf., Porter et al., 2019). To minimise the risk for paraglacial reworking issues, we avoided sampling boulders in the vicinity of these landforms." |
| | | We also discuss landform stability and the influence of the height of the sampled boulders in Sect. 6.1. |
| **Figure 3** | I recommend the authors give readers some better geographic context for the samples collected, especially given the high-resolution basemap here. Please consider adding dots or some sort of markers to the figure (with labels) for each sample collected. I recognize that the authors more or less did this on Figure 7 but it would be helpful in this zoomed in image. Moreover, the moraine delineations are a little complicated and confusing simply as outlines using the same color for the lines. I recommend coloring each moraine with differing shades of light, transparent fill or something like this so readers can more easily distinguish moraine boundaries. | We added dots and labels in the revised figure. We used different colours for the moraines and added a transect in panel (b) to improve the clarity in Fig. 3. |
| **Figure 4** | As previously stated, I am unsure what the purpose is of including every core collected from the bog if they are ultimately not used in the study. I recommend either removing the cores from the figure, or if there is relevant data from multiple cores, include that data in the paper to help corroborate the radiocarbon results from the single (I am assuming?) core. At the very least, the authors need to identify which of the 13 cores on this figure was sampled for radiocarbon dating because I cannot easily tell from the figure. It might even be helpful (if possible at this scale) to put stars or some sort of marker for the relative depths of sample collections for radiocarbon and IRSL. | None of the cores sampled by Lang et al. (1984) was ultimately used for this study. As we only discuss the palyonological data from core "5" in Lang et al. (1984) in further detail, we discarded the Fig. 4 in the original submission and marked the coring site "5" of Lang et al. (1984) in Fig. 3. |

| Line, Figure, or Table | Reviewer comment | Authors' reply |
|---|---|---|
| **Figure 6** | A general comment on the approach to generating the age-depth model shown in this figure. Why did the authors not include the IRSL ages in the age-depth model? If they are not used in the age-depth model, I am unsure why they are even included in the study. In fact, if the lowermost IRSL age is considered in the age-depth model, it might impact the modeled independent age, at least in respect to the uncertainty in the modeled independent age assignment. I feel this is important for the authors to reconcile, especially if they are concerned with leaning too heavily on just one independent dating method (Lines 33-37). If OxCal cannot accommodate IRSL ages in the age-depth model, I recommend the authors use different software like BACON to generate an updated age-depth model that incorporates the IRSL ages.

Second, could the authors somehow make it a little more obvious in the figure that the tephra layer is hypothesized specifically as the Laacher See Tephra? I got a little confused here. | Thanks for the suggestion regarding the age-depth model. We included both the $^{14}$C and the IRSL ages in the model. The error of the basal age turned out to be slightly lower.

We marked the tephra in Fig. 5 as "Laacher See Tephra". |
| **Tables 1 and 3** | Stylistically, I would recommend the authors combine these tables, I am not sure what the purpose is of separating this information. In fact, table 3 comes after figure 6 in the text so readers see the age-depth model before they even see the raw radiocarbon dates and calibrated ages. | The tables were combined in the revised manuscript. |

| Line, Figure, or Table | Reviewer comment | Authors' reply |
|---|---|---|
| **Table 6** | I am not sure why this information needs to be separate from the information in table 5. Moreover, I am unsure why the authors did not report information for the samples they elected to remove from the dataset. I recommend combining the two tables. I also recommend the authors move this combined table up in the text closer to the paragraph in line 280. As is, I had to scroll back and forth several times between the table and the relevant text while reading.

Here and in table 5, based on the information provided, I recalculated $^{10}$Be concentrations (and I commend the authors for providing sufficient data to do so), but they are not identical to the concentrations provided. For example, the first sample in table 5 (FS-01a), the authors report a concentration of 134500 at/g but my calculations for that sample were 137938 at/g, approximately 2.5% higher. All other reported concentrations are lower than my recalculations at roughly the same percentage. Except FS-03a, which was somehow 10% lower than my calculation. Because this is a production rate calibration and has important implications for calculating exposure ages elsewhere, I strongly encourage the authors to reaffirm their reported concentrations and/or if my calculations are correct, update the tables and the entire manuscript accordingly.

As a final point, I am not sure how useful the 'normalized concentrations' column is. These reported concentrations may be normalized for shielding and thickness, but they are not scaled down to SLHL so one still cannot compare 'apples to apples'. I recommend the authors make the full effort to normalize concentrations by including the scaling factor as well as the shielding and thickness corrections and report the completely normalized values. | Tables 5 and 6 were combined in the revised manuscript (Table 4).

As mentioned above, we carefully double-checked the presented Beryllium-10 concentrations. See the attached table for further details.

The reported concentrations were scaled down to SLHL for suitable comparison. Fig. 6 in the revised manuscript shows the normalised Beryllium-10 concentrations with respect to the error-weighted mean Beryllium-10 concentration. See Table 3 for fully normalised Beryllium-10 concentrations (at sea-level and high latitudes). |

| Line, Figure, or Table | Reviewer comment | Authors' reply |
|---|---|---|
| **Additional Figure 1** | Building off the fully normalized concentrations that I feel should be reported for every sample measured (even the one with a low AMS current), I recommend that the authors make an additional figure to graphically display the normalized concentrations. My preference would be for the authors to make normal probability density functions for each sample and a summed pdf (e.g., 'camelplot') so readers can see the normalized concentrations in the context of each other to quickly assess the distribution, but I leave that up to the authors how they want to graphically display normalized concentrations. | We added an additional figure to the manuscript (Fig. 6) showing the normalised Beryllium-10 concentration with respect to the error-weighted mean $^{10}$Be concentration. |
| **Additional Figure 2** | Because the authors are identifying issues with shielding of cosmogenic production, a commonly adopted approach to mitigate some of these issues is by selecting only the largest boulders (higher likelihood of being wind-swept of snow, less likely to have been exhumed post-depositionally or significantly covered by soil/vegetation, etc.), so I recommend the authors consider adding a plot of normalized concentrations versus boulder height. If there is a trend, that might support some of the conclusions drawn by the authors and/or highlighted in this review. | The study site is located in a sheltered position, as a dense forest composed of Norway spruce, beech, and silver fir covers the study site. Sampling large boulders would therefore not help to mitigate the issue of snow shielding. However, we agree that selecting the largest boulders would allow for mitigating post-depositional and post-stabilisation issues. We plotted the boulder height versus the normalised $^{10}$Be concentrations to check whether there is any trend. Figure 8a reveals the absence of a clear trend ($R^2$ = 0.01, $p$ = 0.83). The lack of a clear relationship between these factors supports the idea that other factors (e.g., individual exposure histories) explain the variations in normalised $^{10}$Be concentrations. |

| Line, Figure, or Table | Reviewer comment | Authors' reply |
|---|---|---|
| **Line 74** | You surveyed and sampled 3 moraines, correct? Fix please. Could say something like "the bog is situated stratigraphically between some of the moraines" if that is correct. | We reformulated the sentence as follows:

"We chose the Feldsee Cirque (8.0 °E, 47.9 °N WGS 1984 coordinate reference system) because (i) we observed multiple large, quartz-bearing boulders on two well-preserved moraines and because (ii) a bog, the Feldsee Bog, is situated in the tongue basin of the former glacier whose sediments are stratigraphically younger than these ice-marginal moraines (Lang, 2005; Hofmann and Konold, 2023)."

We hope that it is clear that we included $^{10}$Be concentrations in moraine-boulder surfaces at two ice-marginal position in the calibration dataset. |
| **Line 242-244** | I am confused by this paragraph. You collected samples from FS-03 and FS-02, and then one sample from FS-01, which is the moraine that dams the lake, correct? As written, it makes it seem like you collected more than one sample on FS-01, which I think is not true. I recommend rewriting this paragraph and including the total number of samples collected per moraine (perhaps in parentheses). | We rewrote the paragraph as follows:

"For establishing the BFPR, we collected surface-rock samples (Table 3) from (i) six gneiss boulders on the moraine at position FS-03 and (ii) four gneiss boulders on the ice-marginal moraine at position FS-02. We also sampled the surface of the FS-01a gneiss boulder on the moraine at position FS-01a for age calculations. However, we did not include the sample in the calibration dataset, as moraine formation might have post-dated the onset of deposition of lake sediments at the FSM coring site (Fig. 3b)."

For clarity, we added the ice-marginal positions to Table 3 and 4. |

| Line, Figure, or Table | Reviewer comment | Authors' reply |
|---|---|---|
| **Line 250-252** | Here is where I think you could inject a little more discussion on the morphology/stability of the moraines themselves. Are they mostly matrix supported and susceptible to degradation, is local seismicity an issue, etc. | Thanks for this remark. We added information on the additional information on the moraine's morphology and stability:

"Since the study of Tomkins et al. (2021) demonstrated that landform stability mainly influences the scatter in age distributions from moraines (and thus in 10Be concentrations), only well-embedded boulders were selected to avoid underestimated $^{10}$Be concentrations due to boulder rotating as well as post-depositional and post-stabilisation exhumation. As the moraine at position FS-03 consisted of clast-supported diamicts and some parts of the moraine were solely composed of boulders, identifying stable and large boulders proved to be straightforward. The same was true for the moraine at position FS-02 although this landform consisted of matrix-supported diamicts. Identifying large and stable boulders on the moraine at position FS-01 turned out to be difficult, as this landform consisted of matrix-rich diamicts and since the moraine exhibited only a few boulders. We thus only sampled one large boulder. Hofmann and Konold (2023) mapped a kettle on the proximal side of the moraine at position FS-03 and on the moraine at position FS-02 in the centre of the Feldsee Bog (Fig. 3), pointing to paraglacial reworking and delayed moraine stabilisation (cf., Porter et al., 2019). To minimise the risk for paraglacial reworking issues, we avoided sampling boulders in the vicinity of these landforms." |

| Line, Figure, or Table | Reviewer comment | Authors' reply |
|---|---|---|
| **Line 305** | There are more potential factors that get integrated into a 'baseline' production rate, e.g., glacial isostatic adjustment, atmospheric redistribution, etc. that are elegantly accounted for with reference production rate calibrations. It might be worth mentioning these other factors as well. | We agree that we should have mentioned these factors in the manuscript. We reformulated the sentence as follows: "Following the approach in a previous calibration study (Fenton et al., 2011), a "baseline" production rate was first calculated, i.e., a production rate that accounts for the site-specific bias induced by snow cover, vegetation cover, soil cover, and postdepositional weathering and by other factors, such as changes in atmospheric circulation." |
| **Line 307** | Just curious, how do the resulting reference production rates compare between using CREp and the online exposure age calculator? Are they virtually identical? | The production rates are similar. However, the production rate calculated with the online calculators formerly known as the CRONUS-Earth online calculator came with a larger uncertainty (CREp: $3.64\pm0.11$ atoms $g^{-1}$ quartz $a^{-1}$, CRONUS-Earth: $3.65\pm0.20$ atoms $g^{-1}$ quartz $a^{-1}$). Note that we comment on the production rate derived with the online calculators formerly known as the CRONUS-Earth online calculator in the results section of the paper. See also Table 6. |
| **Line 320** | I am not sure how appropriate it would be to use the erosion rate estimated from Reuther, 2007. The erosion rate is certainly environmentally controlled, but it is also controlled by the lithology – density, age, grain size, etc. Unless the authors specify that the bedrock at their Black Forest site is of a similar lithology, age, density, grain size, etc. to the site in the referenced paper, I feel it would be difficult to assess the validity of using this erosion rate | As we newly identified a protruding quartz vein on the FS-02b boulder, we were able to calculate a site-specific weathering rate (0.06 cm ka$^{-1}$). This weathering rate was based on the basal age of the lake sediments at the FSM coring site and the height of the quartz vein. |
| **Line 328-329** | Might be a sentence/spelling error in this sentence. | Exactly. This was a typo. |

| Line, Figure, or Table | Reviewer comment | Authors' reply |
|---|---|---|
| **Line 465** | Just to confirm, the $^{10}$Be concentrations reported in table 5 and 6 are the blank corrected concentrations, right? The text is slightly vague here. I would recommend explicitly stating that "values reported in the table are blank corrected" so there is no ambiguity. | We have accordingly revised the manuscript. We hope that it is clear that we only refer to blank-corrected concentrations. |
| **Line 610-611** | I think it is a useful finding that there were no protruding quartz veins in the sampled boulders, unlike what was observed in Reuther (2007). To me, this could signify that boulder surfaces were better-preserved and much less weathered than the authors hypothesize. If true, this observation might lend support to the minimally discussed idea of moraine stabilization/exhumation processes impacting cosmogenic nuclide inventories in sampled boulders. I recommend the authors consider and discuss this possibility in more detail. | After sampling, we did not observe quartz veins on the freshly exposed rock surfaces on the boulders. However, we recently went to the field again and carefully inspected the boulders. In contrast to previous field surveys, we noted protruding quartz veins on the FS-02b boulder and on two random moraine boulders at position FS-03. The presence of a quartz vein with a height of 1 cm on the FS-02b boulder suggests that the sampled boulders underwent significant weathering and removal of rock. Note that the weathering corrected production rates reported in Table 5 are based on the site-specific weathering rate. See the previous comments for further details. |
| **Line 652-656** | In terms of data availability, I suggest the authors consider contributing their cosmogenic nuclide measurements to ICE-D (www.ice-d.org) for community discoverability and use. | The calibration dataset will be submitted to ICE-D after the acceptance of this manuscript. |

---

## Author Comment (AC2)

**Reply to the comments of Anonymous Referee #2 to the manuscript entitled 'Local Beryllium-10 production rate for the mid-elevation mountainous regions in Central Europe, deduced from a multi-method study of moraines and lake sediments in the Black Forest'**

Dear reviewer,

We thank you for your thoughtful and critical comments that resulted in considerable improvements to the manuscript. We thoroughly considered all comments and revised the manuscript accordingly. For responses to the comments, see the table below. We hope that the manuscript will be accepted for publication in its revised form.

Thank you very much for your kind consideration.

On the behalf of all co-authors,

Felix Martin Hofmann

| Line, Figure, or Table | Reviewer comment | Authors' reply |
|---|---|---|
| **General comment** | Tables often include incomplete citations. For example, please properly cite the scaling methods, etc. in column headings of Table 7. | The references have been included in the updated table. |
| **Line 12** | Instead of "understanding" maybe "determination" would be better? | The manuscript has accordingly been revised. |
| **Line 13** | Until now might be added before "For the midelevation (Variscan)…" | The manuscript has accordingly been revised. We have reformulated the sentence as follows: "Until now, no calibration site has been available for the mid-elevation mountain ranges of central Europe." |
| **Line 16** | specify that the study uses IRSL, and define its acronym, instead of luminescence dating. | The manuscript has accordingly been revised. |
| **Line 18** | …rate in quartz. (Add "in quartz".) | The manuscript has accordingly been revised. |
| **Line 20** | study site, instead of stud site. | This was a typo. |
| **Line 21** | Seems broadly outside the scope of this manuscript and to the best of my knowledge isn't really addressed in the text. Please reword or remove this line from the abstract. | We have removed this line from the abstract. |
| **Line 27** | Worth citing the CReP calculator Martin et al., 2017 here, too, particularly because it is used later in calculations. | The manuscript has accordingly been revised: "CRE age calculators, such as the cosmic-ray exposure program (CREp; Martin et al., 2017) utilise physical models, such as the Lifton-Sato-Dunai (LSD) scaling scheme (Lifton et al., 2014), to extrapolate 10Be production rates at calibration sites to sampling sites." |
| **Line 29 and throughout text** | Please consider using production-rate calibration site, or calibration site, instead of reference site | "production rate reference site" has been replaced by "calibration site" throughout the manuscript. |
| **Line 30** | "At independently dated reference sites…" and yet only one (Claude et al., 2014) is cited. Please consider adding more references and associated citations | The manuscript has accordingly been revised. |
| **Line 31** | Determination of the rate. | We have accordingly revised the manuscript. |
| **Line 31** | Cosmogenic nuclide production-rate calibration instead of Geological calibration. | "geological calibration" has been replaced by "geological 10Be production-rate-calibration" |

| Line, Figure, or Table | Reviewer comment | Authors' reply |
|---|---|---|
| **Line 33** | several authors, yet only one is cited. Please add appropriate references/citations. | We have inserted the following references: Small and Fabel, 2016a, b; Lowe et al., 2019. |
| **Line 35** | involving multiple, independent dating methods. | The manuscript has accordingly been revised. |
| **Line 36** | the resulting cosmogenic nuclide production rate might be... | The manuscript has accordingly been revised. |
| **Line 39** | such as the CRONUS-Earth and CRONUS-EU projects (Cosmic-Ray prOduced NUclide Systematics) | The manuscript has accordingly been revised. |
| **Lines 39-54** | global canonical Be-10 production rate is mentioned in the abstract, but the value and associated scaling method are neither provide in the abstract nor in this paragraph, which seems to be introducing the background to Be-10 production rates. Ranges of rates produced at European calibration sites could be added and all associated references with citations could be included, rather than directing the reader to read the references in Martin et al. (2017). | We have included the range of European production rates and the global mean production rate. We also added the production rates to the abstract. |
| **Line 46** | more strongly, rather than stronger | The manuscript has accordingly been revised. |
| **Line 48** | may also differ from those (rates) at production-rate calibration sites | The manuscript has accordingly been revised. |
| **Lines 59 – 62** | run-on sentence. Please break into shorter sentences. | We have reformulated the sentence as follows: "As previously discussed (e.g., Hofmann et al., 2020, 2023; Hofmann, 2023), there is an urgent need for dating the onset of retreat from their Late Pleistocene maximum positions. Clarifying this issue would help to evaluate the hypothesis that the Alps shielded the ice caps and glaciers from humid air masses from the Mediterranean Sea during the last major advance of piedmont lobe glaciers in the forelands of the Alps (at around 25 ka; e.g., Gaar et al., 2019)." |

| Line, Figure, or Table | Reviewer comment | Authors' reply |
|---|---|---|
| **Lines 69 and 70** | Obtaining… this sentence reads out of scope, out of place in this paper. | As mentioned in the manuscript, Be-10 cosmic-ray exposure dating is the key method for age determination of moraines in the mid-elevation mountains of central Europe. Therefore, the choice of the production rate has a strong influence on the ages and on the palaeoclimatic interpretation of the age datasets. |
| **Line 73** | Because instead of since. | The manuscript has accordingly been revised. |
| **Line 74** | two well-preserved moraines | The manuscript has accordingly been revised. |
| **Line 76** | accumulated, in situ cosmogenic… | The manuscript has accordingly been revised. |
| **Line 76** | in quartz from moraine-boulder surfaces | The manuscript has accordingly been revised. |
| **Line 77** | Radiocarbon and IRSL dating techniques were used to date layers in sediment cores. | We have reformulated the sentence as follows: "Obtaining sediment cores from the FSM coring site ("FSM" stands for Feldseemoor, the German name of the bog) on a buried moraine, radiocarbon dating of macrofossils, IRSL dating, and establishing an age-depth model with the 14C ages, the IRSL ages, and the age of a cryptotephra allowed us to derive a minimum age for ice-free conditions at the bog." |
| **Line 77** | Reading through the paper, I thought dating was only used on the FSM borehole. | This is correct: "Obtaining sediment cores from the FSM coring site ("FSM" stands for Feldseemoor, the German name of the bog) on a buried moraine…" |
| **Line 78** | a minimum radiocarbon(?) age | We obtained a **modelled age** by interrogating 14C ages, IRSL ages, and the age of a cryptotephra in a Bayesian approach, i.e., an age-depth model in Oxcal. |

| Line, Figure, or Table | Reviewer comment | Authors' reply |
|---|---|---|
| **Line 78-79** | IRSL dating was used as a separate dating technique which independently verifies the sequence of radiocarbon ages in the FSM core. (Or something similar). | We undertook IRSL dating as an additional line of evidence for establishing the age-depth model: "Obtaining sediment cores from the FSM coring site ("FSM" stands for Feldseemoor, the German name of the bog) on a buried moraine, radiocarbon dating of macrofossils, **IRSL dating**, and establishing an age-depth model with the 14C ages, the IRSL ages, and the age of a cryptotephra allowed us to derive a minimum age for ice-free conditions at the bog." |
| **Line 81** | We propose that calibrating the regional production rate finally offers... | The manuscript has accordingly been revised. |
| **Line 82** | I'm not sure this is accurate, the evaluation of other authors' correction factors. The case to do this with a minimum radiocarbon age (for calibration) and the BF sites own set of vegetation, forestation, snow cover, and weathering/erosion issues seems weak at best. | We have removed this sentence from the introduction. |
| **Line 85** | Paragraph needs a topic sentence to introduce the details that are coming. Figure 1 should also be mentioned in the first few lines. | The manuscript has accordingly been revised: "The study site, the Feldsee Cirque, is located in the southern part of the Black Forest in SW Germany (Fig. 1). The Feldsee Cirque is situated about 2 km ESE of Feldberg (1493 m a.s.l.), the highest summit of the Black Forest. Due to the high abundance of glacial landforms (cf., Liehl, 1982; Metz and Saurer, 2012; Hofmann and Konold, 2023), it is a key site for Pleistocene glaciations of the Black Forest. The cirque has attracted glacio-geomorphological and geological research for almost two centuries (Walchner, 1846; Ramsay 1862; Lang et al., 1984; Schreiner, 1990; Hofmann and Konold, 2023)." |
| **Line 86** | specify what "it" is in the sentence that begins "It is situated east..." | The manuscript has accordingly been revised: "The Feldsee Cirque is situated about 2 km ESE of Feldberg (1493 m a.s.l.), the highest summit of the Black Forest." |
| **Line 87** | Feldsee, a moraine-dammed lake up to 33 m deep | The manuscript has accordingly been revised. |
| **Line 93** | The dominant(?) lithology in the study area is a quartz-bearing basement rock of the... | See the reply to the next comment. |

| Line, Figure, or Table | Reviewer comment | Authors' reply |
|---|---|---|
| **Lines 94-97** | Starting with "With the denudation..." and ending with "Mezozoic sedimentary rock." Is this information relevant to this study? The reader only needs to know what rocks are present that could have been incorporated into glacial and/or lake/bog deposits. | We have reformulated the paragraph on the pre-Quaternary geological/geomorphological evolution of the study site as follows: "Quartz-bearing rock of the Variscan basement (age: 380—290 Ma; Geyer et al., 2011), i.e., flaser gneiss, migmatite, porphyry, and paragneiss dominates the study area (LGRB, 2013). In addition, quartz-rich porphyry outcrops on the cirque's western headwall (LGRB, 2023). Since denudation from about 50 Ma onwards (Eberle et al., 2023) has led to the complete removal of the Permian, Triassic, and Jurassic sedimentary rock on the Variscan basement (Wimmenauer et al., 1990), glacial sediments at the study site (mainly till) only originate from quartz-rich rock of Variscan age (Schreiner, 1990)." |
| **Figure 1** | Include a citation/reference in the figure for the assumed late Pleistocene maximum ice extent. Location of the study area in the southern Black Forest, DE... | We have reformulated the figure caption as follows: "Figure 1: Topographical map of the southern Black Forest showing the assumed maximum ice extent during the Late Pleistocene (Hemmerle et al., 2016), ice divides (Hemmerle et al., 2016), and outlet glacier names according to the nomenclature of Hofmann et al. (2020). See NASA Jet Propulsion Laboratory (2013) for information on the digital elevation model (DEM) in the background. The inset map shows the location of the Black Forest in Germany." |
| **Line 104** | "...was repeatedly glaciated." Citations or References? | We have inserted appropriate references: Liehl, 1982; Metz and Saurer, 2012; Hemmerle et al., 2016 |
| **Line 106** | refer to Figure 1. | We have accordingly revised the manuscript. |
| **Lines 109 to 111** | there are three ranges of ages presented and only two valleys mentioned. What does "respectively" refer to in this sentence? | We agree that "respectively" is not needed. |

| Line, Figure, or Table | Reviewer comment | Authors' reply |
|---|---|---|
| **Figure 2** | Oblique aerial photograph of the study area (study sites?)…Shown are the headwall of the Feld see Cirque…Label the prominent moraine in the figure.The semicircular moraine is also represented by a dotted line (as is the prominent moraine). Use a different symbol? | We have revised the figure caption and labeled the moraines in the figure. |
| **Line 119** | This needs a topic sentence that moves the reader from Pleistocene glacial times into the present day and leading them to the connection you're trying to make. | Thanks for this remark. We have added the following sentence: "Although the Feldsee cirque glacier at study site area has long since disappeared, snow cover still plays an important role today." |
| **Lines 119 – 121** | Run-on sentence. Please break into shorter sentences. | We have shortened the sentence. |
| **Line 120** | Why is this specific 30-yr period used? Why not a longer period of time? Why not a period of time that goes farther back in time? | Data on mean temperature, average precipitation,…, in Germany should be given for a 30-yr period following the recommendations of the World Meteorological Organization (https://library.wmo.int/viewer/ 55797?medianame=1203_en_#page= 1&viewer=picture&o=bookmark&n=0&q=). Following the guidelines of the World Meteorological Organization, this period (1961-1990 CE) is usually selected, as the climate during this period is only partly affected by ongoing climate change. |
| **Line 121 and 122** | I read the sentence that starts with "Snowfall…" and found myself asking "And…?" after reading it. Why is this important? What does it indicate relative to your study? | We have added the following sentence to the study site section: "As it will be discussed below, the Feldsee Cirque thus is a challenging site where seasonal snow cover might have considerably slowed down the accumulation of in situ produced $^{10}Be$ in moraine-boulder surfaces." |
| **Line 126** | refer to Figure 3? | The manuscript has accordingly been revised. |
| **Figure 3** | Don't abbreviate position to pos. What does FSM stand for? It should also be in the legend. | The figure and the caption have accordingly been revised. |
| **Line 155** | Why is only coring site 5 mentioned? Isn't there the same sequence in each core? | Lang et al. (1984) only identified the apparent tephra at their coring site 5. We removed all other cores from the manuscript, as palyonological data is only available for the coring site 5. |

| Line, Figure, or Table | Reviewer comment | Authors' reply |
|---|---|---|
| **Line 159** | Make it clear to the reader that this ash in this core was not dated. It has been hypothesized to correlate with the Laacher See Tephra which has a reported age of... | We have reformulated the sentence as follows "At one coring site (coring site "5" in Lang, 2005), these authors observed a distinct greyish layer at a depth of about 8.1 m below the ground surface (Fig. 4b). Lang et al. (1984) speculated that this layer is the Laacher See Tephra, having a reported age of 13006±9 cal. a BP (Reinig et al., 2021)." |
| **Line 168** | Specify the FSM core in this section? | We refer to the FSM coring site at the beginning of Sect. 3: "To the best knowledge of the authors, the cores obtained by Lang et al. (1984) do, unfortunately, not exist anymore. We thus obtained sediment cores at the FSM coring site during fieldwork in 2021 CE." |
| **Line 174** | ...content of organic matter in layers? | We have reformulated the sentence as follows: "Sediment samples were obtained from the cores, dried, and loss-on-ignition (LOI) analyses (cf., Heiri et al., 2001) were undertaken." |
| **Figure 4** | Sedimentary successions at 13 coring sites of Lang (2005). Instead of redrawn from, write Modified after Lang (2005). | We have removed the figure, since we only comment on the sediment cores obtained from the coring site "5". The Lateglacial part of the cores is shown in Fig. 4b. |
| **Line 182** | describe method used to determine water content. | We have added the following sentence: "Weighing the samples prior to drying and before the LOI analyses allowed for the determination of the sediments' water content. " |

| Line, Figure, or Table | Reviewer comment | Authors' reply |
|---|---|---|
| **Line 184** | Individual conversion factors should be listed (in a table?) for each of the 1 m core lengths. Percent adjustment should be specified (values?). Brief explanation of decompacted depths is needed. | We have included the conversion factors in the results section: "Vibracoring at the FSM coring site allowed for obtaining sediment cores with a total length of 8 m. Borehole FSM recovered the sedimentary succession of the Feldsee Bog and the uppermost 0.32 m of the partly buried moraine at position FS-02 (Fig. 5). The percentage of sediment recovery increased from 67% between a depth of 4 and 5 m to 81% between a depth of 5 and 6 m. Decompaction was thus undertaken with correction factors of 1.49 (4-5 m) and 1.23 (5-6 m). Beryllium-10 concentrations in a total of 10 moraine-boulder surfaces were successfully determined, allowing for production rate calibration." We have included further information on the decompaction procedure in the methods section: "During opening of the cores in the lab, we noted that all sediment cores were shorter than the penetrated depth and, thus, core shortening must have occurred during vibracoring. Generally, core shortening is one of the main limitations of this technique (cf., Glew et al., 2001). As mentioned by Glew et al (2001), sediments with a higher water content are generally more prone to compaction. We assumed that only the clayey and silty lake sediments in the cores (water content: 18-85%) were affected by shortening and not the stratigraphically older diamicts (water content: 15-17%). Following Glew et al. (2001), we assumed that the sediments in the cores were progressively thinned down-core, i.e., equally affected by compaction. Individual conversion factors were computed for every one-metre-long sediment core which then allowed for adjusting the thickness of the lithostratigraphic units to the penetrated depth." |
| **Line 187** | This paragraph/section needs a topic sentence. | We have added a topic sentence: "To numerically date the sediments at the coring site, radiocarbon dating of macrofossils was undertaken." |

| Line, Figure, or Table | Reviewer comment | Authors' reply |
|---|---|---|
| **Line 187** | Samples of macrofossils were hand picked… "See Supplement for photos of macrofossil samples (Figures x - y)." should be its own sentence. | The manuscript has accordingly been revised. |
| **Line 195** | It is unclear what "were assumed to be in correct stratigraphical order" means. | We have reformulated the sentence as follows: "The *P_Sequence* function was selected in Oxcal, as the 14C ages of the macrofossils and the IRSL ages of sediment samples were expected to increase with depth" |
| **Table 1** | Consider combining tables 1 and 3. There is redundancy between them. | Tables 1 and 3 have been combined. |
| **Sect. 3.3** | should be IRSL dating. | We have renamed the section: **3.3 IRSL dating** |
| **Line 202** | This section needs a topic sentence. | We have added the following topic sentence: "To cross-check the radiocarbon ages, seven sediment samples from two of the cores (depth: 4-6 m below the ground surface) were sampled for luminescence dating under subdued red-light, with two further samples taken to account for potential dose-rate inhomogeneity due to the complex stratigraphy (FSM-D1 and FSM-D2; Table 2)." |
| **Line 202** | Seven samples were obtained from the core… Specify which core and the depth at which each sample was collected. List sample names or refer reader to a table with them listed. | We have moved the table with the results of IRSL dating to Sect. 3.3. |
| **Line 212** | For all samples, a standard IRSL protocol was used. Please add a reference/citation. | We have inserted the appropriate reference: "For all samples, a standard IRSL protocol (modified from Preusser, 2003) was applied." |

| Line, Figure, or Table | Reviewer comment | Authors' reply |
|---|---|---|
| **Lines 214-215** | the word latter is used twice, making the second latter unclear. What does it refer to? | We have reformulated the sentences as follows:
"This protocol comprised a preheat to 250 °C for 60 s and IRSL stimulation at 50 °C for 100 s (IR-50). For fine grains, a post-IR (pIR) IRSL protocol was additionally tested to potentially overcome the need for fading correction. This protocol involved a preheat to 250 °C for 60 s, IRSL stimulation at 50 °C for 100 s, and a second stimulation at 225 °C for 100 s (pIR)." |
| **Line 217** | first mention of OSL. Should that also be in the section title? | No, that is not neccessary. Optically stimulated luminescence measurements on quartz revealed no suitable signal: "Optically stimulated luminescence measurements on quartz revealed no suitable signal, similar to reports on other directly bedrock derived samples (e.g., Preusser et al., 2006) and experience from the nearby Upper Rhine Graben (Preusser et al., 2016, 2021). Therefore, potassium feldspar was selected as dosimeter." |
| **Line 234** | "uranium" is not capitalized. This small error occurs several times in this section. | The manuscript has accordingly been revised. |
| **Line 242** | Edit to: We collected surface-rock samples... | The manuscript has accordingly been revised. |
| **Line 244** | We also sampled the surface of one boulder | The manuscript has accordingly been revised. |
| **Table 2** | Add a column indicating which moraine or site from which each of the samples was collected. Column for measured sample density? Column for dip angle and azimuth for each sloping surface? Significant figures in the topographic shielding factor? | The ice-marginal positions have been added to the table. Strike and dip of the sampling surfaces are given in the detailed sample documentation in the supplement. We have added the following sentence to the section: "See the detailed sample documentation for strike and dip of the sampling surfaces, measured with a geological compass.". |
| **Line 254** | Indicate that the angle and azimuth of sloping surfaces was collected and used to add to the total shielding correction. | We have reformulated the sentence as follows: "Therefore, the ArcGIS toolbox of Li (2018) was chosen for shielding factor calculations, considering both self-shielding of dipping surfaces and shielding by topographical obstructions around the sampling sites." |

| Line, Figure, or Table | Reviewer comment | Authors' reply |
|---|---|---|
| **Line 261** | mention that you account for deep-forest shielding in your total shielding. | We did not account for deep-forest shielding to be able to compute the "baseline" production rate. |
| **Lines 271 and 273** | These concentrations of acid are very strong. Are these not typically diluted for treatment of mineral separates? | The acids were not diluted during the preparation of the samples. |
| **Line 290** | local, unscaled production rates? SLHL production rates? Please be specific. | We have reformulated the sentence as follows: "The calibration of the spallogenic 10Be SLHL Black Forest production rate followed the workflow of Martin et al. (2017, their Fig. 3)." |
| **Line 293** | mean latitude, longitude, and elevation? Why not use sample specific latitude, longitude, and elevation? | This is the standard workflow of CREp described in Martin et al. (2017). |
| **Line 294** | Peirce. Incomplete citation? | We have reformulated the sentence as follows: "Following the guidelines of Ross (2003), the 10Be concentrations were subsequently evaluated with Peirce's criterion (Peirce, 1852), and a weighted 10Be concentration was computed after the exclusion of outliers." |
| **Line 297** | uncertainties are 1-sigma? 2-sigma? | We have reformulated the sentence as follows: "As recommended by Martin et al. (2017), the standard error of the weighted mean 10Be concentration (calculated with $1\sigma$ uncertainties of the 10Be concentrations) was multiplied with $\sqrt{MSWD}$ to obtain the uncertainty of the average 10Be concentration." |
| **Line 300** | Give the actual age and uncertainty of the modeled radiocarbon age against which production rates are calibrated. | We have included the modelled basal age. |

| Line, Figure, or Table | Reviewer comment | Authors' reply |
|---|---|---|
| **Line 302** | It appears only two scaling methods were applied, rather than "all scaling schemes and…" Please be specific in language. Lm and LSD were applied, if I understand correctly. | We have reformulated the sentence as follows: "The spallogenic SLHL 10Be BFPR in quartz was computed for the scaling schemes in CREp, i.e., time-dependent Lal/Stone (Nishiizumi et al., 1989; Lal, 1991; Stone, 2000; Balco et al., 2008) and LSD (Lifton et al., 2014) scaling, and all geomagnetic databases in CREp, i.e., the atmospheric 10Be-based virtual dipole moment (VDP; Muscheler et al., 2005, and references therein), the LSD framework (Lifton et al., 2014), and the Lifton 2016 VDM (Lifton, 2016, and references therein)." |
| **Line 354** | Hofmann et al. (2022) recently recalculated CRE ages…Why is this important here? It feels out of place in this manuscript. Is there additional text you could add to explain to the reader why this is relevant? | We have reformulated the beginning of the section as follows: "To assess the impact of the newly calibrated BFPR on CRE ages, CRE ages, internal (analytical) uncertainties, and external uncertainties (i.e., analytical uncertainties plus the error of the 10Be production rate added in quadrature) for the sampled moraine-boulder surfaces were calculated with the Chironico landslide spallogenic SLHL production rate and the BFPR." We have also renamed the whole section: **3.6 Assessment of the impact of the new production rate** |
| **Line 356** | To assess the effect of the choice of production rate… Do you refer to the choice of production rate in this study? The study of Hofmann et al. (2022)? It's not clear why this is relevant to this calibration study. | See the reply to the previous comment. |
| **Line 359** | Were the production rates not also scaled using LSD? | The spallogenic SLHL Black Forest production rate was also scaled with LSD. See Table 5 for further details. |
| **Lines 365 – 378** | It seems like circular reasoning to me to calibration a production rate from moraine boulder Be-10 concentrations and then use that same production rate to calculate exposure ages | The idea here was to perform a sensitivity test. How much do the ages shift if they are calculated with the new production rate. If we had chosen another age dataset from the southern Black Forest, the age difference would have been the same. |

| Line, Figure, or Table | Reviewer comment | Authors' reply |
|---|---|---|
| **Line 381** | This section needs a topic sentence. Does this section give descriptions for each core collected? Is it just for the FSM core? If the latter, why only the FSM and not the others? | We only refer to the cores obtained from the FSM coring site. We have included the following paragraph at the beginning of Sect. 5: "Vibracoring at the FSM coring site allowed for obtaining sediment cores with a total length of 8 m. Borehole FSM recovered the sedimentary succession of the Feldsee Bog and the uppermost 0.32 m of the partly buried moraine at position FS-02 (Fig. 5). The percentage of sediment recovery increased from 67% between a depth of 4 and 5 m to 81% between a depth of 5 and 6 m. Decompaction was thus undertaken with correction factors of 1.49 (4-5 m) and 1.23 (5-6 m). A total of 11 moraine-boulder surfaces was sampled to determine 10Be concentrations for production rate calibration." |
| **Lines 381 – 384** | This paragraph is very unclear and confusing. Context? | See the reply to the previous comment. |
| **Line 396** | Maybe move this paragraph to the beginning of this section? | In Sect. 5.1 we first describe the results of logging and LOI analyses. In the subsequent paragraph (starting with "The succession of FSM (Fig. 5) reflects the glacial-postglacial transition of the study area. The FSM borehole...") we interpret the results. We argue that the results and the interpretation should be discussed in two separate paragraphs. |
| **Figure 6** | Caption mention cores. Is this figure only for the FSM core? The * and ** should be a complete sentence or two in the caption. Do you mean dark read lines instead of curves? List sample numbers for IRSL ages. Hard to tell triangles(?) from circles(?). I was unable to see any symbol/line that is light blue. | We have revised the figure caption as follows: "Sediment sequence at the FSM coring site, calibrated ages (95% ranges of calibration), IRSL ages, the age-depth model, and LOI. Photos of the sediment cores from the Feldsee Bog were acquired with the methodology of Gegg and Gegg (2023). 95% ranges of the modelled ages are marked with dashed lines. The solid line represents the mean modelled age. Agreement indices (A) for individual ages are given in parentheses." |
| **Line 416** | Needs a topic sentence. Also, specify if the radiocarbon ages are from the FSM core. | We have added the following topic sentence: "A total of nine macrofossils in sediment cores from the FSM coring site was radiocarbon dated." |

| Line, Figure, or Table | Reviewer comment | Authors' reply |
|---|---|---|
| **Table 4** | is this related to the FSM borehole? Please specify. | We have reformulated the phrase as follows: "Table 4 summarises the luminescence data for the FSM coring site." |
| **Line 468** | Paragraph needs topic sentence. | We have added the following topic sentence: "Calibrating the spallogenic BFPR with the 10Be concentrations in the FS-02a, FS-02b, FS-02c, FS-03a, FS-03c, FS-03d, FS-03e and FS-03f moraine-boulder surfaces resulted in SLHL production rates between 3.61±0.11 and 3.65±0.11 10Be g-1 quartz a-1 for the different scaling schemes and geomagnetic databases in CREp (Table 5)." |
| **Line 470** | Consider using "samples were scaled" rather than "normalized". Also, table 6 doesn't exemplify this. The scaling factors are listed in the table as are the scaled Be-10 concentrations. | We have corrected the concentrations for topographic shielding and the sample thickness and adjusted the concentrations to SLHL. See Table 4. |
| **Table 6** | Please consider adding a column with LSD scaling factors. Specify the other column is the "Lm" scaling factors. | We have included the scaling factors for both scaling schemes in Table 4. |
| **Table 7** | Are these global, arithmetical means? Error-weighted means? SLHL values? How many samples contribute to these? Please specify. Consider combining tables 6 and 7. | At the beginning of Sect. 5.6, we state that "Calibrating the spallogenic BFPR in CREp (Martin et al., 2017) with the 10Be concentrations in the FS-02a, FS-02b, FS-02c, FS-03a, FS-03c, FS-03d, FS-03e and FS-03f moraine-boulder surfaces resulted in SLHL production rates between 3.61±0.11 and 3.95±0.12 10Be g-1 quartz a-1 (Table 5) for the different scaling schemes and geomagnetic databases in CREp (Martin et al., 2017)." |

| Line, Figure, or Table | Reviewer comment | Authors' reply |
|---|---|---|
| **Sections 6.2 and 6.3** | **Sections 6.2 and 6.3 don't seem relevant, in my opinion, to the scope of this calibration paper.** | We argue that these sections are highly relevant for our calibration cover. In Sect. 6.2, we invalidate a previous approach to correct for snow cover and post-depositional weathering in the mid-elevation mountain ranges of central Europe. We therefore propose that the newly calibrated production rate should be utilised for age determination rather than production rates at calibration sites outside this region. In Sect. 6.3, we discuss the newly calibrated production rate in the European context. We argue that it should be mandatory to compare a new production rate with previously established production rates. |
| **Section 6.4** | The title of section 6.4 would be better posed more like a statement rather than as a question. | We have accordingly revised the title of the section: "Explanations for the anomalously low BFPR" |